# *CNTN5⁻/⁺* or *EHMT2⁻/⁺* human iPSC-derived neurons from individuals with autism develop hyperactive neuronal networks

Eric Deneault[1,2], Muhammad Faheem[1,2], Sean H White[3], Deivid C Rodrigues[4], Song Sun[5,6,7], Wei Wei[4], Alina Piekna[4], Tadeo Thompson[4], Jennifer L Howe[1,2], Leon Chalil[3], Vickie Kwan[3], Susan Walker[1,2], Peter Pasceri[4], Frederick P Roth[5,6,7,8,9], Ryan KC Yuen[1,2], Karun K Singh[3]*, James Ellis[7]*, Stephen W Scherer[1,2,7,10]*

[1]Genetics & Genome Biology Program, The Hospital for Sick Children, Toronto, Canada; [2]The Centre for Applied Genomics, The Hospital for Sick Children, Toronto, Canada; [3]Department of Biochemistry and Biomedical Sciences, Stem Cell and Cancer Research Institute, McMaster University, Hamilton, Canada; [4]Developmental & Stem Cell Biology Program, The Hospital for Sick Children, Toronto, Canada; [5]Lunenfeld-Tanenbaum Research Institute, Mount Sinai Hospital, Toronto, Canada; [6]The Donnelly Centre, University of Toronto, Toronto, Canada; [7]Department of Molecular Genetics, University of Toronto, Toronto, Canada; [8]Department of Computer Science, University of Toronto, Toronto, Canada; [9]Canadian Institute for Advanced Research (CIFAR), Toronto, Canada; [10]McLaughlin Centre, University of Toronto, Toronto, Canada

*For correspondence:
singhk2@mcmaster.ca (KKS);
jellis@sickkids.ca (JE);
stephen.scherer@sickkids.ca (SWS)

**Abstract** Induced pluripotent stem cell (iPSC)-derived neurons are increasingly used to model Autism Spectrum Disorder (ASD), which is clinically and genetically heterogeneous. To study the complex relationship of penetrant and weaker polygenic risk variants to ASD, 'isogenic' iPSC-derived neurons are critical. We developed a set of procedures to control for heterogeneity in reprogramming and differentiation, and generated 53 different iPSC-derived glutamatergic neuronal lines from 25 participants from 12 unrelated families with ASD. Heterozygous de novo and rare-inherited presumed-damaging variants were characterized in ASD risk genes/loci. Combinations of putative etiologic variants (*GLI3/KIF21A* or *EHMT2/UBE2I*) in separate families were modeled. We used a multi-electrode array, with patch-clamp recordings, to determine a reproducible synaptic phenotype in 25% of the individuals with ASD (other relevant data on the remaining lines was collected). Our most compelling new results revealed a consistent spontaneous network hyperactivity in neurons deficient for *CNTN5* or *EHMT2.* The biobank of iPSC-derived neurons and accompanying genomic data are available to accelerate ASD research.
**Editorial note:** This article has been through an editorial process in which authors decide how to respond to the issues raised during peer review. The Reviewing Editor's assessment is that all the issues have been addressed (see decision letter).
DOI: https://doi.org/10.7554/eLife.40092.001

## Introduction

The past two decades of research has determined autism spectrum disorders (ASD) to be clinically (*Fernandez and Scherer, 2017*; *Jones and Lord, 2013*; *Mahdi et al., 2018*) and genetically (*De Rubeis et al., 2014*; *Gilman et al., 2011*; *Pinto et al., 2014*; *Tammimies et al., 2015*; *C Yuen*

*et al., 2017*) heterogeneous. Phenotypically, the fifth edition of the Diagnostic and Statistical Manual of Mental Disorders (DSM-5) combines autistic disorder, Asperger disorder, childhood disintegrative disorder and pervasive developmental disorder not otherwise specified into the single grouping of ASD (*DSM-5, 2013*). There are also syndromic forms of ASD (*Carter and Scherer, 2013*), and now more than 100 other disorders carrying different names (*Betancur, 2011*), that in a proportion of subjects can also present the necessary symptoms for an ASD diagnosis.

From the perspective of genetics, heritability estimates and family studies definitely demonstrate genes to be involved (*Ronald and Hoekstra, 2011*). Single high-penetrance genes and copy number variation (CNV)-affected loci, have now been implicated as *bona fide* autism-susceptibility (or risk) genes, although none of them show specificity for ASD alone (*Malhotra and Sebat, 2012*). These genetic alterations are rare in the population (<1% population frequency), and in some individuals, combinations of rare genetic variants affecting different genes can be involved (*Devlin and Scherer, 2012*), including more complex structural alterations of chromosomes (*Brandler et al., 2018*; *Marshall et al., 2008*). Recent research studying common genetic variants indicates that polygenic contributors may be involved, and these can also influence the clinical severity of rare penetrant variants in ASD risk genes (*Weiner et al., 2017*).

Nearly 1000 putative ASD risk loci are catalogued, with ~100 already being used in the clinical diagnostic setting (*Hoang et al., 2018a*; *Winden et al., 2018*). There are some genotype-phenotype associations emerging, including general trends considering medical complications and IQ (*Bishop et al., 2017*; *Sanders et al., 2015*; *Tammimies et al., 2015*), sibling variability depending on the ASD gene variant they carry (*Yuen et al., 2015*), and lower adaptive ability in those carrying variants compared to affected siblings without the same genetic change (*C Yuen et al., 2017*). Many of the ASD risk genes identified are connected into gene networks including those involved in synaptic transmission, transcriptional regulation, and RNA processing functions (*Bourgeron, 2015*; *De Rubeis et al., 2014*; *Geschwind and State, 2015*; *Pinto et al., 2014*; *Sahin and Sur, 2015*; *C Yuen et al., 2017*; *Yuen et al., 2016*), with the impacted genes being involved in all of prenatal, region-specific, or broader brain development (*Uddin et al., 2014*). Perhaps, a general unifying theme that is emerging from neurophysiologic studies is an increased ratio of excitation and inhibition in key neural systems that can be perturbed by variants in the ASD risk genes, or by environmental variables affecting the same targets (*Canitano and Pallagrosi, 2017*).

The advent of the induced pluripotent stem cell (iPSC) technology (*Takahashi et al., 2007*; *Yu et al., 2007*), followed by cellular re-programming to forebrain glutamatergic neurons (*Habela et al., 2016*), allows accessible cellular models to be developed for the highly heterogeneous ASD (*Beltrão-Braga and Muotri, 2017*; *Dolmetsch and Geschwind, 2011*; *Durak and Tsai, 2014*; *Karmacharya and Haggarty, 2016*; *Marchetto et al., 2017*; *Yoon et al., 2014*; *Zhang et al., 2013*). Carrying the same precise repertoire of rare and common genetic variants as the donor proband, iPSC-derived neurons represent the best genetic mimic of proband neurons for functional and mechanistic studies. Induced differentiation can be achieved with high efficiency and consistency using transient ectopic expression of the transcription factor NGN2 (*Ho et al., 2016*; *Zhang et al., 2013*), and this has been shown to be useful in diverse phenotyping projects (*Deneault et al., 2018*; *Pak et al., 2015*; *Yi et al., 2016*). Proband-specific iPSC-derived neuronal cells indeed provide a useful model to study disease pathology, and response to drugs, but throughput (both iPSC-derived neurons and phenotyping) is low, with costs still high. As a result, so far, only a few iPSC-derived neuronal lines are typically tested in a single study.

Here, we develop a resource of 53 different iPSC lines derived from 25 individuals with ASD carrying a wide-range of rare genetic variants, and from unaffected family members. We also used clustered regularly interspaced short palindromic repeats (CRISPR) editing (*Jinek et al., 2012*; *Ran et al., 2013*) to create four 'isogenic' pairs of lines with or without mutation, to better assess mutational impacts. Upon differentiation into excitatory neurons, we investigated synaptic and electrophysiological properties using the large-scale multi-electrode array (MEA), as well as more traditional patch-clamp recordings. Numerous interesting associations were observed between the genetic variants and the neuronal phenotypes analyzed. We share our general experiences and the bioresource with the community. We also highlight one of our more robust findings—an increased neuronal activity in glutamatergic neurons deficient in one copy of *CNTN5* or *EHMT2*—which could be responsible for ASD-related phenotypes.

## Results

### Selection and collection of tissue samples for reprogramming

Participants were enrolled in the Autism Speaks MSSNG whole-genome sequencing (WGS) project (*C Yuen et al., 2017*). All ASD and related control-participants were initially consented for WGS and upon return of genetic results, then consented for the iPSC study, using approved protocols through the Research Ethics Board at the Hospital for Sick Children (see Materials and methods section for details) (*Hoang et al., 2018b*). Some families were also examined by whole exome sequencing. The study took place over a 5 year period and used incrementally developing ASD gene lists from the following papers (*Jiang et al., 2013*; *Marshall et al., 2008*; *Tammimies et al., 2015*; *Yuen et al., 2015*) (*Table 1*). These primarily considered data from the Autism Speaks MSSNG project, the Autism Sequencing Consortium (*De Rubeis et al., 2014*), and the Simons Foundation Autism Research Initiative (SFARI) gene list (discussion below). A diversity of different ASD-risk variants was targeted ranging in size from single nucleotide variants (SNV) to an 823 kb CNV (*Figure 1* and *Table 1*; corresponding genomic coordinates in *Supplementary file 1*). Typically, one ASD-affected and one sex-matched unaffected member (control) per family were included (*Figure 1*). In total, 14 ASD-affected and 11 controls participated, of which 21 were males and four were females (*Figure 1* and *Table 1*). Cells from either skin fibroblasts or CD34 +blood cells were collected for reprogramming into iPSCs (*Figure 2A* and *Table 1*).

### Derivation of iPSC lines

Two different viral approaches were used for cell reprogramming. For historical reasons, the first three cell lines in *Table 1*, namely iPSC IDs 19–2, 19-4 and NR3, were reprogrammed using retroviruses expressing *OCT4/POU5F1*, *SOX2*, *KLF4* and *MYC*, and a lentiviral vector that encoded the pluripotency reporter EOS-GFP/Puro$^R$ (*Hotta et al., 2009*). Then, we moved to non-integrative Sendai virus for all the other tested lines (*Table 1*). Emerging iPSC colonies were selected for activated endogenous human pluripotency markers, differentiation potential into three germ layer cells after embryoid body formation in vitro, and normal karyotype (*Figure 2B–D* and *Supplementary file 2*). Two separate pluripotent and karyotypically normal iPSC lines were typically selected per participant for neuronal differentiation and phenotyping experiments (*Table 1*).

### Transient induction of neuronal differentiation

We induced differentiation of newly generated iPSCs into glutamatergic neurons to test their electrophysiological properties (*Figure 2A*). We used the NGN2 ectopic expression approach since highly-enriched populations of glutamatergic neurons can be obtained within a week, and they exhibit robust synaptic activity when co-cultured with glial cells (*Zhang et al., 2013*). Importantly, we determined that this strategy offers highly uniform differentiation levels between cell lines derived from different participants (*Deneault et al., 2018*). This consistency was necessary to perform suitable phenotyping assays such as network electrophysiology recordings of several different lines in the same experimental batch. The resulting glutamatergic neurons were all subjected to electrophysiological phenotyping.

### Multi-Electrode array analysis of iPSC-derived neurons

MEA phenotyping was predominantly used in order to monitor the excitability of several independent cultured neuron populations in parallel, and in an unbiased manner, as we previously adapted with different NGN2-neuron lines (*Deneault et al., 2018*). We sought to determine if any selected ASD-risk variants would interfere with spontaneous spiking and synchronized bursting activity in a whole network of interconnected glutamatergic neurons. We ensured that the duration and amplitude of detected spikes were similar to typical mammalian neurons, that is, action potential widths of around 1–2 milliseconds (ms) and peak amplitudes of approximately 20–150 µV (*Figure 2—figure supplement 1A*). We measured the glutamatergic/GABAergic nature of our cultured neurons produced using NGN2 ectopic expression, which is known to repress GABAergic differentiation at the advantage of glutamatergic (*Roybon et al., 2010*). Mean firing rate (MFR) and network bursting activity were measured upon treatment with different receptor inhibitors. No substantial change was observed after addition of the GABA receptor inhibitor PTX (*Figure 2—figure supplement 1B*),

**Table 1.** List of participants with ASD or unaffected controls, with the genetic variant(s) involved, and the different iPSC lines derived. *The 1 bp deletion in EHMT2 would result in a frameshift 47 codons before the end of the protein and disruption of the stop-codon, potentially leading to the inclusion of a total of 221 incorrect amino acids; more information corresponding to the different genetic variants are presented in *Supplementary file 1*; MZ, monozygotic; Retro, retrovirus; N/A, not available

| Family ID | MSSNG Id | Status | Primary genetic variant(s) | Sex | Age at reprogramming (year) | Cell of origin | Reprogramming method | iPSC ID | Reference |
|---|---|---|---|---|---|---|---|---|---|
| ASD Candidate Gene - CNVs | | | | | | | | | |
| A | 1-0019-002 | Unaffected father | Family and study control | M | 44 | Skin | Retro | 19–2 | *Deneault et al., 2018* |
| | 1-0019-004 | ASD-affected | 16p11.2 deletion/+ | M | 15 | Skin | Retro | 19-4 | *Marshall et al., 2008* |
| B | 3-0368-000 | ASD-affected | *NRXN1* 430 kb deletion/+ | M | 8 | Skin | Retro | NR3 | *Tammimies et al., 2015* |
| C | 1-0262-002 | Unaffected father | Family control | M | 49 | Skin | Sendai | 16K, 16N | — |
| | 1-0262-003 | ASD-affected | *DLGAP2* 791 kb duplication/+ | M | 10 | Skin | Sendai | 15E, 15G | *Marshall et al., 2008* |
| | 1-0262-004 | Affected brother | Family control | M | 14 | Skin | Sendai | 17E, 17G | — |
| D | 1-0582-002 | Unaffected father | Family control | M | 37 | Skin | Sendai | 26E, 26J | — |
| | 1-0582-003 | ASD-affected | *CNTN5* 676 kb deletion | M | 9 | Skin | Sendai | 27H, 27N | N/A |
| E | 7-0058-003 | ASD-affected | *AGBL4* 323 kb deletion/+ | M | 4 | Skin | Sendai | 36O, 36P | N/A |
| ASD Candidate Gene – SNVs | | | | | | | | | |
| F | 2-1305-005 | Unaffected brother | Family control | M | 7 | Skin | Sendai | 21H, 21P | — |
| | 2-1305-003 | ASD-affected | *CAPRIN1* p.Q399X/+ | M | 12 | Skin | Sendai | 20C, 20E, 75G, 75H | *Jiang et al., 2013* |
| G | 2-1186-002 | Unaffected father | Family control | M | 43 | Blood | Sendai | 54E, 54G | — |
| | 2-1186-003 | ASD-affected | *VIP* p.Y73X/+ | M | 12 | Blood | Sendai | 53G, 53H | *Jiang et al., 2013* |
| H | 2-1303-004 | Unaffected brother | Family control | M | 13 | Skin | Sendai | 19A | — |
| | 2-1303-003 | ASD-affected | *ANOS1* p.R423X | M | 19 | Skin | Sendai | 18C, 18E | *Jiang et al., 2013* |
| | 2-1303-003 | Corrected ASD-affected | CRISPR-corrected *ANOS1* p.X423R | M | 19 | Skin | Sendai | 18CW | — |
| I | 1-0273-002 | Unaffected father | Family control | M | 45 | Blood | Sendai | 51C, 51E | — |
| | 1-0273-003 | ASD-affected | *THRA* p.R384C/+ | M | 14 | Blood | Sendai | 52A, 52C | *Yuen et al., 2015* |
| Functional ASD Candidate Genes - SNVs | | | | | | | | | |
| J | 1-0494-005 | Unaffected brother | Family control | M | 12 | Blood | Sendai | 50A, 50B, 50H | — |
| | 1-0494-003 | ASD-affected MZ twin | *SET* c.112 + 1G>C/+ | M | 9 | Blood | Sendai | 48K, 48N | N/A |
| | 1-0494-004 | ASD-affected MZ twin | *SET* c.112 + 1G>C/+ | M | 9 | Blood | Sendai | 49H, 49G | N/A |

*Table 1 continued on next page*

*Table 1 continued*

| Family ID | MSSNG Id | Status | Primary genetic variant(s) | Sex | Age at reprogramming (year) | Cell of origin | Reprogramming method | iPSC ID | Reference |
|---|---|---|---|---|---|---|---|---|---|
| K | 7-0254-001 | Unaffected mother | *GLI3* p.G727R/+ | F | 37 | Blood | Sendai | 64N, 64Q | — |
| | 7-0254-002 | Unaffected father | *GLI3* p.G465R/+ | M | 41 | Blood | Sendai | 63Q, 63T | — |
| | 7-0254-003 | ASD-affected | *GLI3* p.G727R/+, mat *GLI3* pG465R/+, pat *KIF21A* p.R1156G/+ (mosaic 23%) | F | 7 | Blood | Sendai | 62M, 62X | N/A |
| | 7-0254-004 | Affected brother | *GLI3* p.G727R/+ *GLI3* pG465R/+ | M | 9 | Blood | Sendai | 61I, 61K | N/A |
| L | 6-0393-001 | Unaffected mother | Family control | F | 54 | Skin | Sendai | 37E | — |
| | 6-0393-003 | ASD-affected | *\*EHMT2* p.K1164Nfs/+ *UBE2I* p.E78K/+ | F | 18 | Skin | Sendai | 38B, 38E | N/A |

DOI: https://doi.org/10.7554/eLife.40092.005

indicating that GABAergic neurons are not appreciably present in our cultures. However, the MFR was significantly reduced in the presence of the AMPA receptor inhibitor CNQX while unchanged in untreated cells, with a comparable profile across each selected line (*Figure 2—figure supplement 1B*). This further suggests that most of the cultures were composed of glutamatergic neurons, and that our induction protocol was consistent across different cultures. All activity was abolished after addition of the sodium channel blocker TTX (*Figure 2—figure supplement 1B*), indicating that our human neurons were expressing functional sodium channels.

The weighted MFR (wMFR), which represents the MFR per active electrode, was used as a primary read-out for all tested iPSC-derived neurons, at one-week intervals from week 4 to 8 post-NGN2-induction (PNI) (*Figure 3*). To identify a preferred timepoint for this screen, we first pooled the data of all the independent control lines. Since the highest wMFR value for this pool of 'all controls' (~1.8 Hz) was detected at week 6 (*Figure 3A*), we initially used that timepoint to compare the activity of ASD variant and control lines for each family. In two different families, that is, *CNTN5* and *EHMT2*, a significant higher wMFR was recorded in ASD variant neurons at week 6 compared with their corresponding familial control neurons (*Figure 3B*). *EHMT2* had a strikingly increased wMFR at all timepoints, whereas *CNTN5* at other timepoints was equivalent to its controls. We therefore ranked these two genes as high priorities for further study. In contrast, no significant differences were observed for *DLGAP2*, *CAPRIN1*, *SET* or *GLI3* (*Figure 3C*), suggesting that these variants do not differ from control neuronal activity in our MEA assays, and therefore were not studied further. Different dynamics of altered wMFR were observed for *ANOS1* and *VIP* at week 4 (*Figure 3D*), and the *ANOS1* nonsense variant was ranked as an example of a candidate for further study. Conversely, a significant lower wMFR was recorded at weeks 7 and 8 for *THRA* (*Figure 3D*). No unaffected family members were available as controls for the single NR3 line (*NRXN1*) nor for 36O-36P (*AGBL4*), thus they were not chosen for further study. When we compared their values to the pooled values recorded from all the different familial controls available, no difference was found for *NRXN1* and a significantly lower wMFR was observed at weeks 5 and 7 for *AGBL4* (*Figure 3E*).

To explore intra-individual (different lines from the same individual) and inter-individual (different individuals with the same mutation) variability, we plotted all the values obtained from each single well, independent experiment, cell line and individual, at each of the five reading timepoints (*Figure 3F*). Most lines from an individual were not significantly different from each other, and reassuringly low inter-individual variability was observed with different siblings bearing the same mutation(s), for example 48K and 48N versus 49G and 49H (*SET*), or 61I and 61K versus 62M and 62X (*GLI3*), at different timepoints (*Figure 3F*). A few lines showed a significant intra-individual variability, for example lines 52A and 52C (*THRA*) at week 4, or lines 75G and 75H (*CAPRIN1*) at week 8 (*Figure 3F*). We also noted some inter-independent experiment variability for a given line,

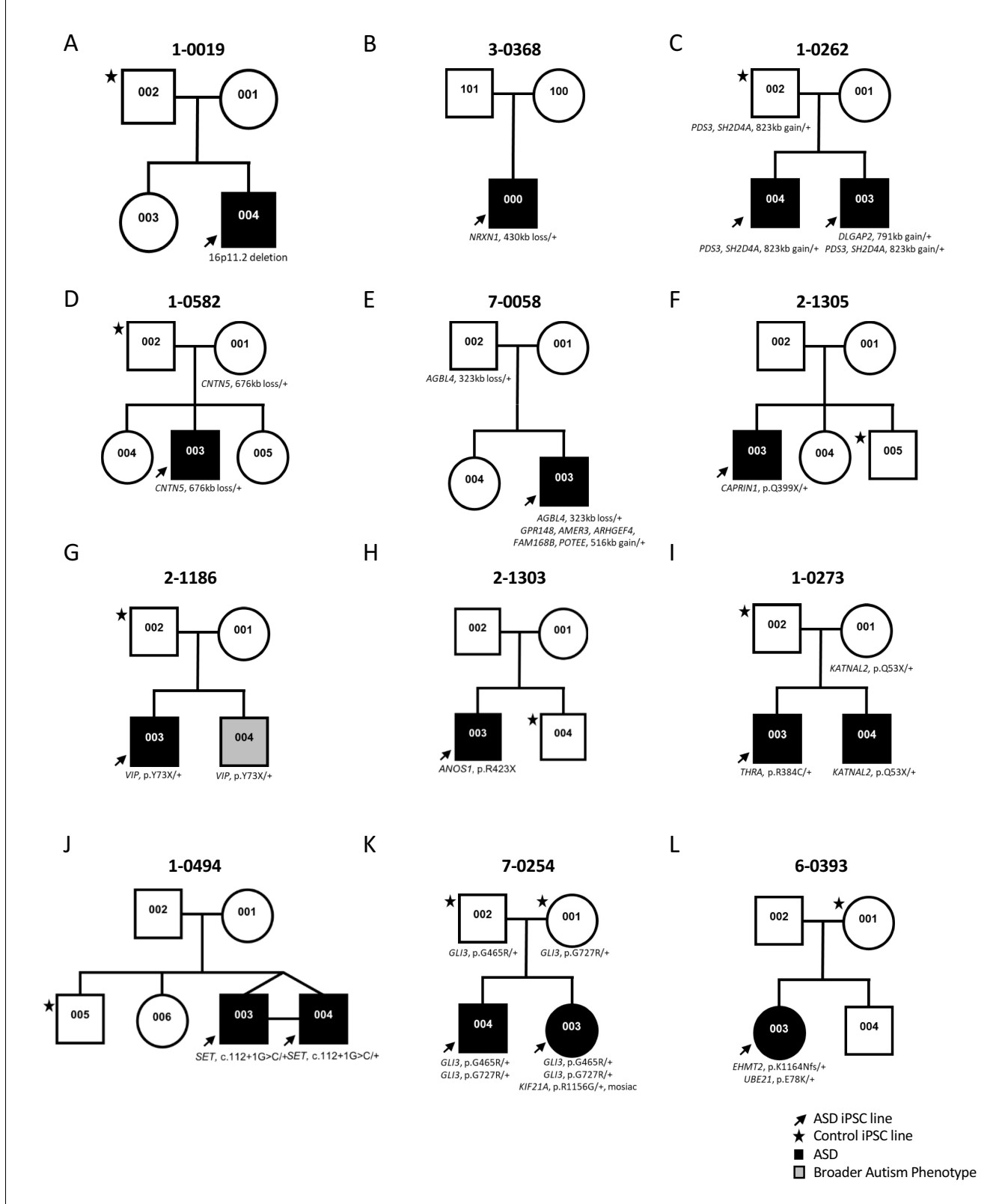

**Figure 1.** Genetic pedigrees of the participant families with identified genetic variants. One ASD-affected (black arrow) and one sex-matched unaffected (black star) members were typically selected for iPSC reprogramming. ASD-affected children are represented with a black box; note that line 1-0019-002 (19-2) in A) was used as a control and was described previously (*Deneault et al., 2018*).
DOI: https://doi.org/10.7554/eLife.40092.002

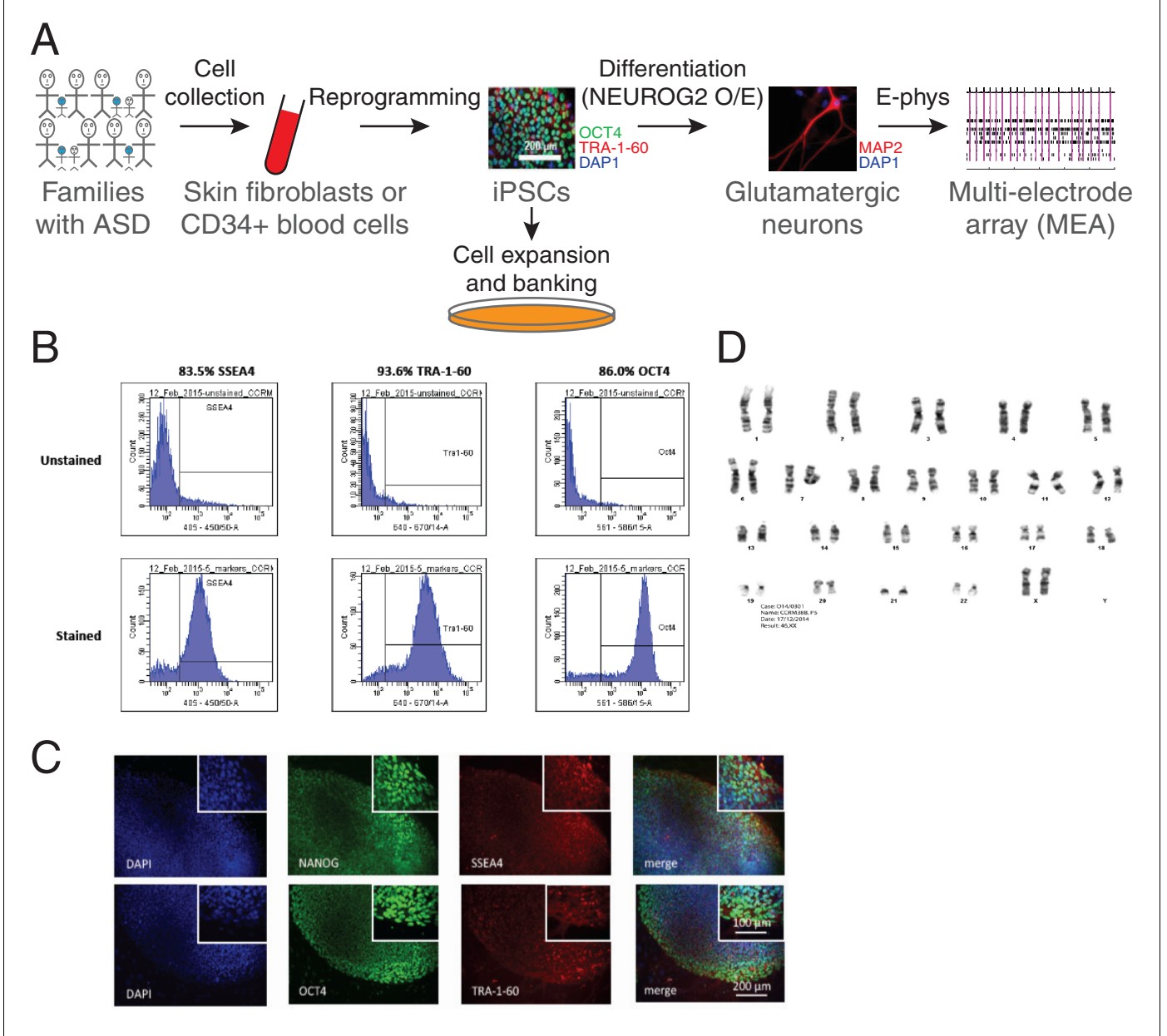

**Figure 2.** Generation of iPSCs and neurons. (**A**) Schematic representation of the experimental procedure to find specific electrophysiological signatures associated with genetic variants of clinical significance to autism spectrum disorder (ASD). Fibroblasts or blood cells were reprogrammed into iPSCs from a cohort of 25 probands and unaffected family members. Differentiation of iPSCs into glutamatergic neurons was achieved with NGN2 7 day transient overexpression, and electrophysiological properties were monitored using a multi-electrode array (MEA) device. (**B**) Flow cytometry and (**C**) Immunohistochemistry revealing expression of the pluripotency markers NANOG, SSEA4, OCT4 and TRA-1–60 in a representative iPSC line. (**D**) Representative normal male karyotype in iPSC; 20 cells were examined.

DOI: https://doi.org/10.7554/eLife.40092.003

The following figure supplement is available for figure 2:

**Figure supplement 1.** Multi-electrode array (MEA) monitoring of iPSC-derived neurons.
DOI: https://doi.org/10.7554/eLife.40092.004

for example line 38E (*EHMT2*) at weeks 4 and 5 (dots with different colours do not overlap in *Figure 3F*). Note that similar profiles were monitored in terms of MFR (*Figure 3—figure supplement 1*), indicating that these differences were not due to having more or less active electrodes in different lines. While consistent activity across lines was generally observed, the presence of

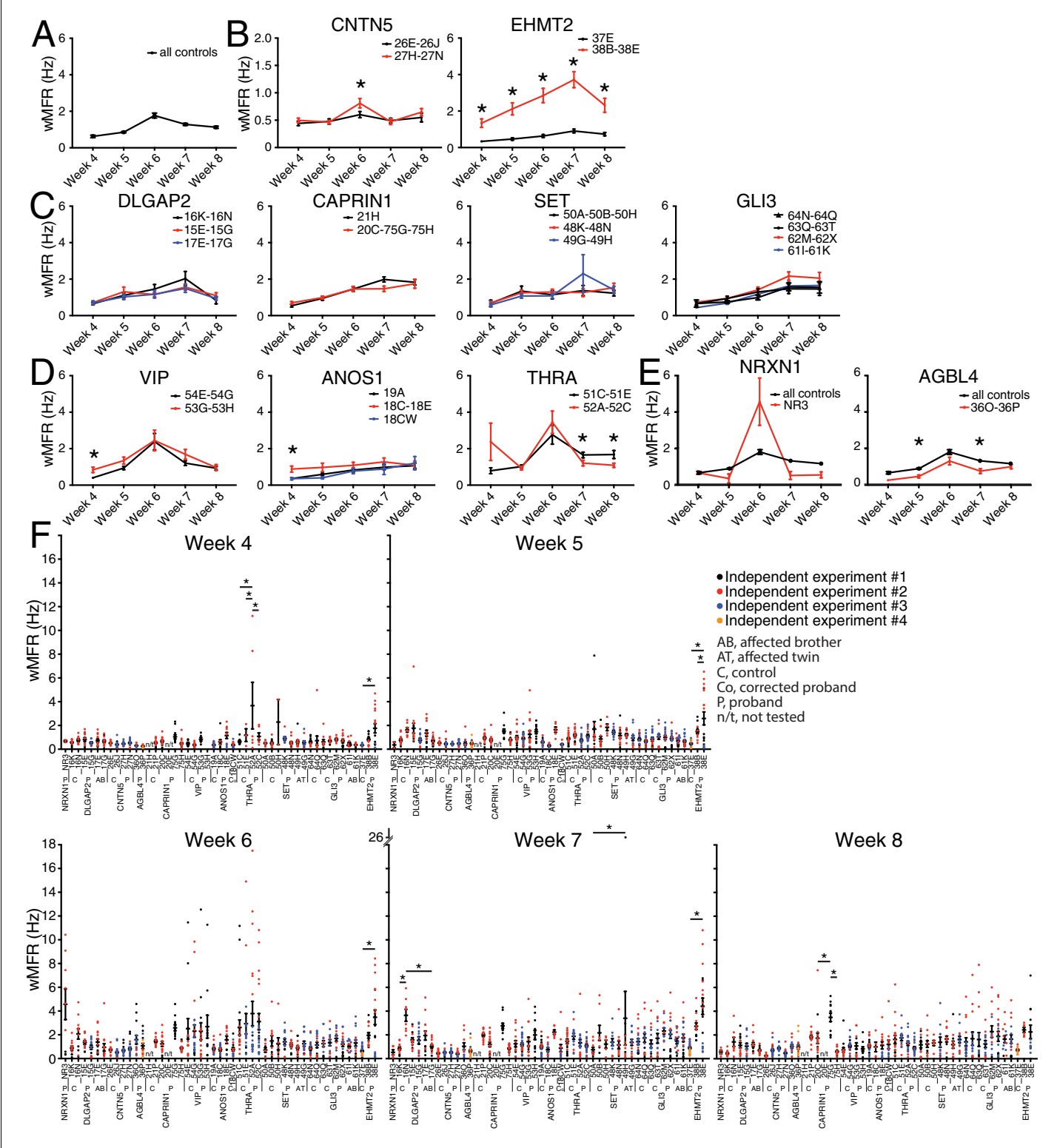

**Figure 3.** Multi-electrode array monitoring of iPSC-derived glutamatergic neurons. (**A–E**) Weighted mean firing rate (wMFR) of pooled cell lines from control and KO neurons for each family from week 4 to 8 PNI. (**F**) Dot plots showing wMFR of each cell line from week 4 to 8 PNI; each dot represents the wMFR of one well, and the color reflects independent experiments. Values are presented as mean ± SEM of several technical and biological replicates, as presented in ***Supplementary file 3***; 'all controls' represents the pool of 311 different control wells from 17 independent experiments;

*Figure 3 continued on next page*

*Figure 3 continued*

*p<0.05 from multiple t test comparison with Holm-Sidak correction (**B**), and without correction (**C–E**), and one-way ANOVA Tukey test pointing to intra- or inter-individual variability per family (**F**).

DOI: https://doi.org/10.7554/eLife.40092.006

The following source data and figure supplements are available for figure 3:

**Source data 1.** Weighted mean firing rate values for each cell line at each timepoint.

DOI: https://doi.org/10.7554/eLife.40092.009

**Figure supplement 1.** Mean firing rates recorded by MEA from iPSC-derived glutamatergic neurons.

DOI: https://doi.org/10.7554/eLife.40092.007

**Figure supplement 1—source data 1.** Mean firing rate values for each cell line at each timepoint.

DOI: https://doi.org/10.7554/eLife.40092.008

variability prompted us to interrogate independent variants created by genome editing of *CNTN5*, *ANOS1* and *EHMT2*.

## *CNTN5* isogenic pair to control for genetic background contribution

To further characterize the heterozygous *CNTN5*-mutant neuron lines 27H and 27N, we first showed a significantly higher network burst frequency at weeks 5, 6 and 8 (*Figure 4A*), indicating a more synchronized neuronal activity across each well. Importantly, CNTN5 protein levels overall were reduced by at least 33% in *CNTN5*$^{-/+}$ neurons (*Figure 4A*, right panel), suggesting that the 676 kb heterozygous loss in *CNTN5* interferes with the production of CNTN5 protein, but also that the non-deleted allele may be more active transcriptionally than in controls.

Unaffected sex-matched family members are genetically similar to their related probands, but still present substantial genetic differences that can contribute to a given phenotype. Isogenic cell pairs represent better control of the genetic background contribution (*Hoffman et al., 2019*). CRISPR editing provides the possibility to engineer such isogenic controls (*Miyaoka et al., 2014*; *Powell et al., 2017*). Since editing large CNVs, such as the 676 kb deletion in *CNTN5*, is currently difficult using existing technology, we elected to introduce a set of nonsense mutations, previously described as 'StopTag' (*Deneault et al., 2018*), to knock out (KO) the expression of this gene in an unrelated iPSC line that was previously generated from a non-ASD and non-carrier individual. This parental line '19–2' was also exploited in similar isogenic KO approaches (*Woodbury-Smith et al., 2017*) (Ross et al., in revision; Zaslavsky et al., in press), allowing assessment in a different and unrelated genetic background. For technical reasons, we targeted exon 5 of the transcript ENST00000524871.5 of *CNTN5* in order to disrupt its expression. A heterozygous iPSC line was isolated to better mimic the heterozygous status of the *CNTN5* deletion in the proband lines 27H and 27N. Intriguingly, the new isogenic iPSC-derived neuron line 19–2-*CNTN5*$^{StopTag/+}$ did not show significant differences in terms of wMFR or network burst frequency at week 6 (*Figure 4B*). However, the wMFR of line 19–2 increased up to nearly 3 Hz at week 8 (*Figure 4B*) while the *CNTN5* family controls stayed around 0.5 Hz (*Figure 3B*). In this context of a more active cell line, we extended the recordings until week 11, and the hyperactive wMFR of 19–2-*CNTN5*$^{StopTag/+}$ was only evident from week 10 (*Figure 4B*). Moreover, CNTN5 protein levels were clearly decreased in this isogenic mutant line (*Figure 4B*, right panel), implying that StopTag insertion efficiently disrupted gene expression. These results indicate that loss of CNTN5 function is responsible for increased neuronal activity in vitro.

## Repair of *ANOS1* rescues defective membrane currents

In a complementary approach to minimize the confounding effect of genetic background from familial and unrelated controls, and its impact on phenotype, we sought to edit our proband-specific variants using CRISPR in order to create matching isogenic controls. We prioritized the nonsense variant R423X found in *ANOS1* in participant 2-1303-003 and successfully corrected the corresponding iPSC line 18C (*Figure 5A–C*). Indeed, after detecting 7% edited cells using droplet digital PCR (ddPCR) in well G08 in the primary 96-well culture plate post-nucleofection, two subsequent limiting-dilution enrichment steps were necessary to isolate a 100% corrected iPSC line (*Figure 5B*).

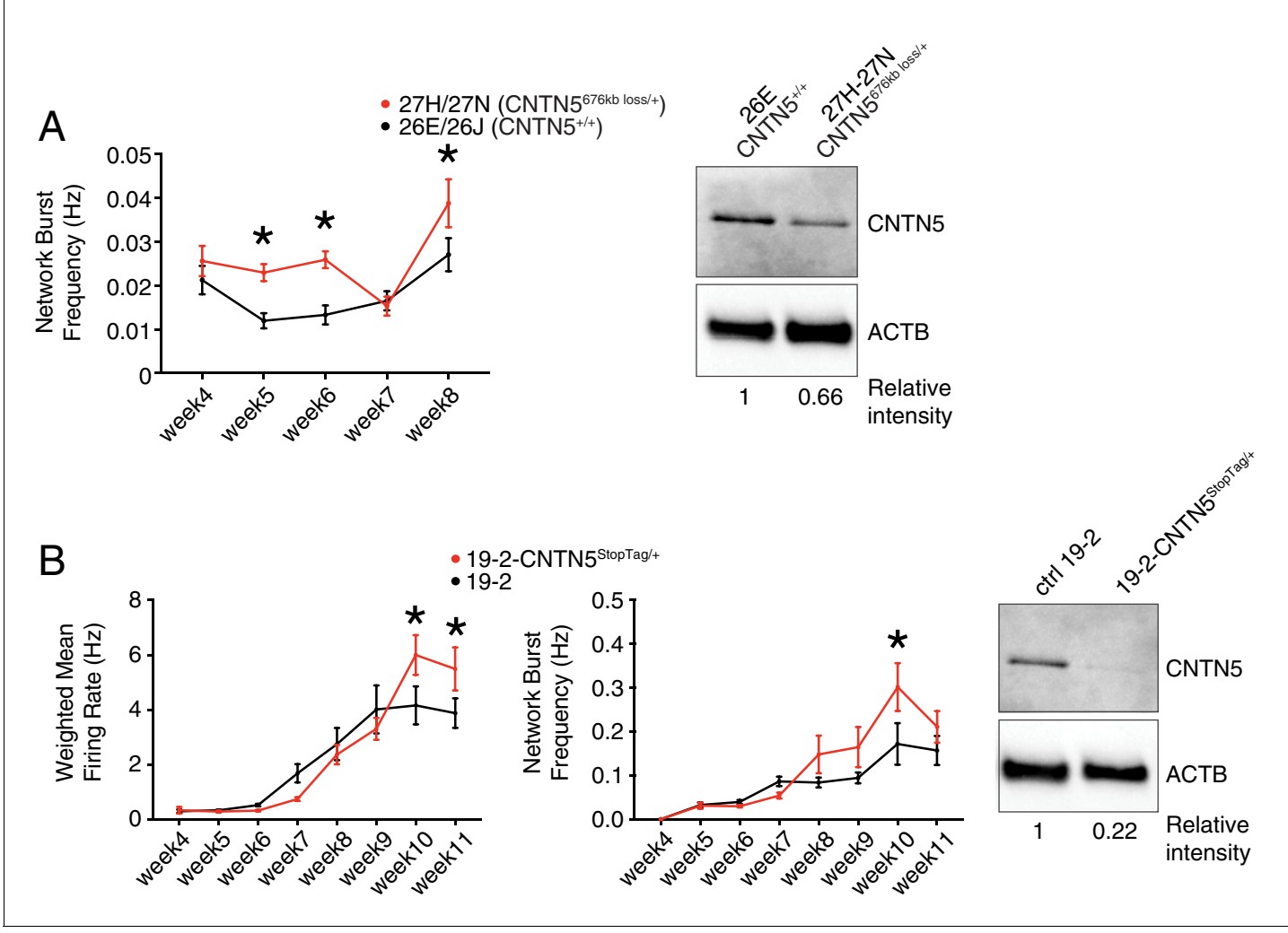

**Figure 4.** Validation of CNTN5-mutant neuron hyperactivity. (**A**) The network burst frequency was recorded from the CNTN5 family from weeks 4–8 PNI, with corresponding protein levels by western blot on the right panel; *p<0.05 from multiple t test comparison with Holm-Sidak correction at weeks 6 and 8. (**B**) Both wMFR and network burst frequency were recorded from the 19–2-CNTN5 isogenic pair from weeks 4–11 PNI, with protein levels. The iPSC IDs and genotypes are indicated above each graph; values are presented as mean ± SEM of different lines per participant, and of several technical and biological replicates, as presented in *Supplementary file 3*; actin beta (ACTB) was used as a loading control for the western blots and the relative intensity of each band is indicated below the blots; *p<0.05 from multiple t test comparison with Holm-Sidak correction.

DOI: https://doi.org/10.7554/eLife.40092.010

The following source data is available for figure 4:

**Source data 1.** Multielectrode array values for familial CNTN5 lines.
DOI: https://doi.org/10.7554/eLife.40092.011

**Source data 2.** Multielectrode array values for isogenic CNTN5 lines.
DOI: https://doi.org/10.7554/eLife.40092.012

Sanger sequencing confirmed the properly corrected genomic DNA sequence (*Figure 5C*). This newly corrected line was named '18CW' (see iPSC line ID '18CW' in *Table 1* and *Figure 5C*).

The CRISPR-corrected line 18CW exhibited a significant difference in wMFR compared with its isogenic counterpart 18C at 4 week, and no difference from the familial control line 19A (*Figure 3D*). Moreover, the availability of such isogenic set prompted us to explore more detailed electrophysiological properties using patch-clamp recordings of single neurons in order to reveal any phenotype not detected using MEA. While the advantage of MEA experiments is that continuous live monitoring of neural activity can be measured over multiple weeks, we used patch-clamp electrophysiology on NGN2 neurons between days 21–25 PNI, which provides robust recordings to

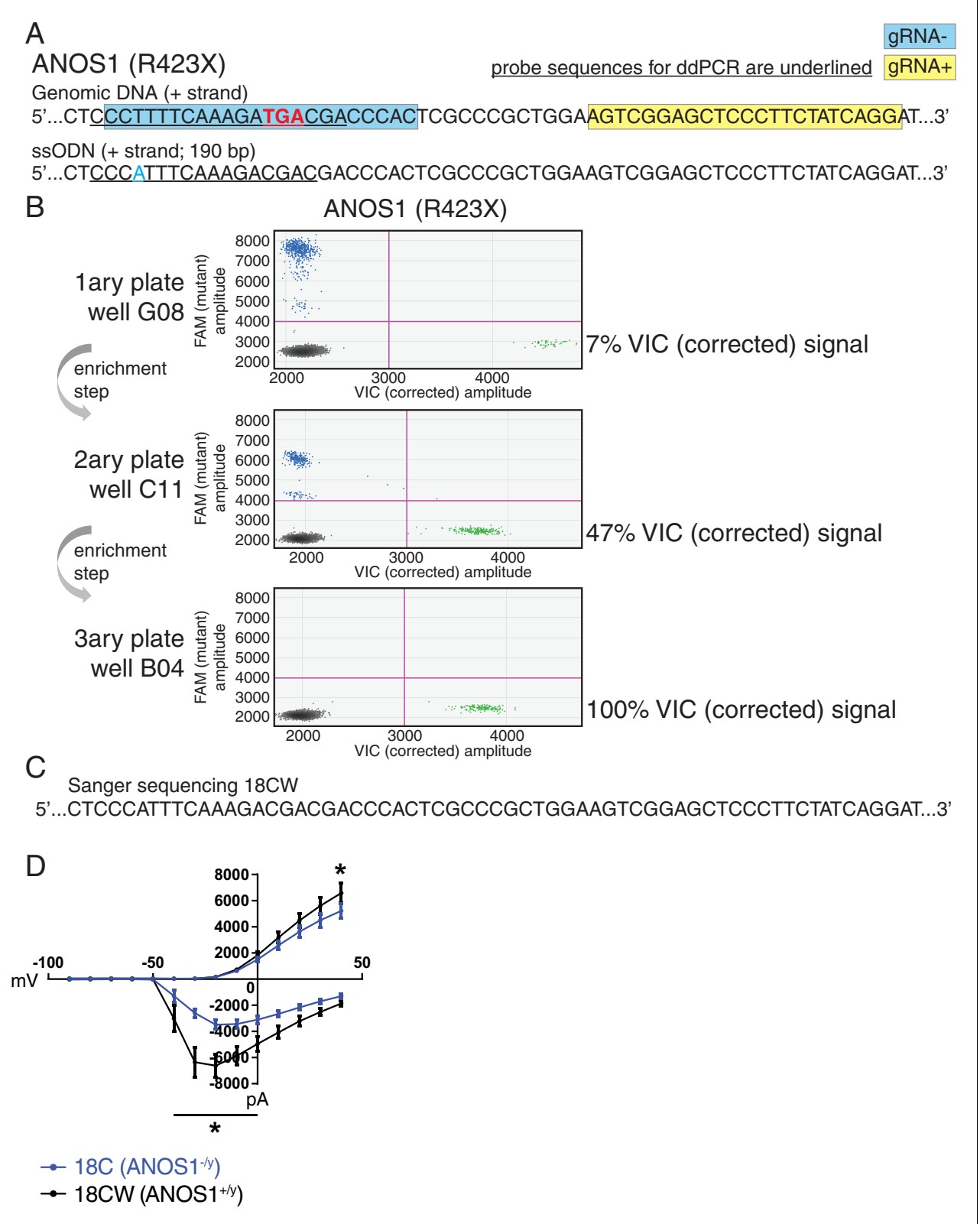

**Figure 5.** Correction of point mutation in ANOS1 in iPSCs using CRISPR editing. (**A**) Design of gRNAs, ssODNs and ddPCR probes for correction of R423X in ANOS1; one sgRNA for each genomic DNA strand, that is, gRNA- in blue and gRNA +in yellow, was devised in close proximity for the double-nicking system using Cas9D10A; the non-sense mutations in ANOS1 is depicted in bold red; a silent mutation was introduced in ssODN (in blue) for ddPCR probe (underlined) specificity and to prevent nicking. (**B**) ddPCR absolute quantification coupled with two consecutive limiting-dilution

*Figure 5 continued on next page*

*Figure 5 continued*

enrichment steps were necessary to isolate a 100% corrected line, that is, 100% VIC signal. (**C**) Sanger sequencing confirmed proper correction of non-sense mutation R423X in line 18C back to wt; this newly corrected line was named 18CW. (**D**) Outward and inward membrane current detected by patch-clamp recordings; total number of recorded neurons was 15 for both 18C and 18CW; values are presented as mean ± SEM of three independent differentiation experiments, recorded at day 21–25 PNI. *p<0.05 from multiple t test comparison with Holm-Sidak correction.

DOI: https://doi.org/10.7554/eLife.40092.013

The following source data is available for figure 5:

**Source data 1.** Inward/outward current values for familial ANOS1 lines.

DOI: https://doi.org/10.7554/eLife.40092.014

detect phenotypes, as shown in previous studies (*Yi et al., 2016*). Furthermore, the increased density of neuronal processes appearing beyond 4 weeks PNI can preclude consistent clean patch-clamp recordings, but this is not an issue with MEA. Using this protocol, we detected significantly lower outward membrane current at 40 mV in the mutant line 18C compared to its isogenic control 18CW (*Figure 5D*). A significantly higher inward current was also observed in mutant neurons between −40 and 0 mV (*Figure 5D*). No overt off-target mutations were detectable using our previously-described WGS strategy (*Deneault et al., 2018*). These results indicate that *ANOS1*-null iPSC-derived glutamatergic neurons present abnormal sodium and potassium membrane currents that might contribute to ASD development. Notably, these observations underline that some specific electrophysiological phenotypes at the single cell level, for example membrane currents, may not be captured when using MEA monitoring at the cell population level.

## Neuronal hyperactivity in *EHMT2/UBE2I* Complex-Variant neurons

Lines 38B and 38E from participant 6-0393-003 carry two ASD-relevant variants; a de novo missense E78K in *UBE2I* and a de novo frameshift variant K1164Nfs in *EHMT2* (*Figure 1L* and *Table 1*). MEA recordings showed a significantly higher wMFR (*Figure 3B*) and network burst frequency (*Figure 6A*) from week 4 to 8 PNI compared to their related control line 37E. Interestingly, the profile of the wMFR curve (*Figure 3B*) was similar to that of the MFR curve (*Figure 3—figure supplement 1A*), indicating that cell survival or expansion is not a major contributor to the difference observed in neuronal activity. To ensure that this hyperactivity was synaptic and not only intrinsic to the neurons, we performed patch-clamp recordings, at day 21–25 PNI to avoid the increased density of neuronal processes that impacts the ability to obtain clean recordings, as stated previously. Intrinsic properties, for example capacitance and resistance, did not vary significantly (*Figure 6B*), indicating comparable maturity levels between lines 37E and 38E. While spontaneous excitatory post-synaptic current (sEPSC) amplitude was unchanged, sEPSC frequency was significantly higher in mutant neurons compared to controls (*Figure 6B*). These observations suggest that a potential loss-of-function of *UBE2I* and/or *EHMT2* is involved in ASD-related neuronal dysfunction.

## Evidence of functional impact of *EHMT2*, but not *UBE2I* variants

Since our attempts to edit the variants E78K in *UBE2I* and K1164Nfs in *EHMT2* had not been successful, we sought to determine the potential contribution of E78K in *UBE2I* to the observed synaptic hyperactivity. To estimate the damaging potential of this missense variant on the function of UBE2I protein, we utilized a *Saccharomyces cerevisiae* complementation assay that was previously developed as a validated surrogate genetic system to predict the pathogenicity of diverse human variants (*Sun et al., 2016*). In this assay, lethality of a temperature-sensitive allele of the yeast *UBC9* gene (ortholog of human *UBE2I*) is rescued by expressing a functional version of human UBE2I. Several missense variants in *UBE2I* have been accurately predicted as deleterious at conserved positions, or benign at other positions (*Zhang et al., 2017*). Therefore, we used this complementation assay to test the consequence of our variant E78K, and found no effect of this variant on the function of human UBE2I (*Figure 6—figure supplement 1*). Because these results disfavor involvement of the *UBE2I* variant E78K in the neuronal hyperactivity observed in *Figures 3B,F and 6A–B*, we excluded *UBE2I* from subsequent experiments and further explored a potential causal link between *EHMT2* and synaptic activity.

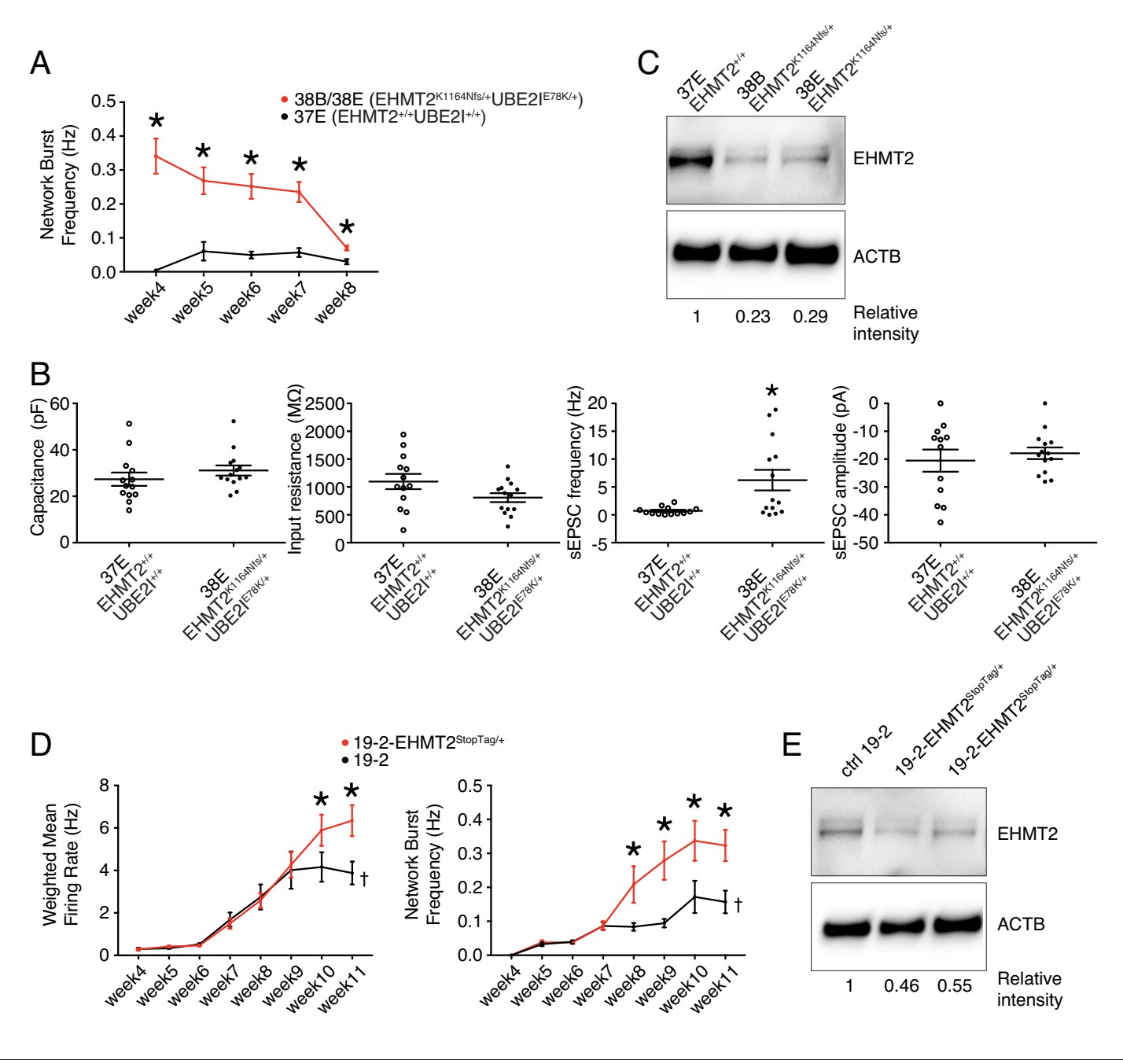

**Figure 6.** Electrophysiological and protein level variations in EHMT2-deficient neurons. (**A**) Network burst frequency was recorded using MEA from the EHMT2/UBE2I family from weeks 4–8 PNI; values are presented as mean ± SEM of several technical and biological replicates, as presented in *Supplementary file 3*; *p<0.05 from multiple t test comparison with Holm-Sidak correction. (**B**) Patch-clamp recordings of two selected lines, that is, 37E (control) and 38E (mutant); values are presented as mean ± SEM of 14 different neurons from two independent differentiation experiments; *p<0.05 from from unpaired t test two-tailed. (**C**) Western blot showing a decrease in EHMT2 protein levels in mutant neurons (38B and 38E) compared to their respective control neurons (37E). (**D**) MEA recordings of the isogenic pair 19–2 and 19–2-EHMT2^StopTag/+ iPSC-derived neurons from weeks 4–11 PNI; values are presented as mean ± SEM of eight different wells for each three independent differentiation experiments; †note that the same data for control 19–2 was used in *Figure 4B* since it was generated within the same experiments, that is, plates 26, 33 and 37 (see *Supplementary file 3*); *p<0.05 from multiple t test comparison with Holm-Sidak correction at week 11 (weighted mean firing rate) and weeks 9–11 (network burst frequency). (**E**) Western blot showing a decrease in EHMT2 protein levels in mutant neurons 19–2-EHMT2^StopTag/+ compared to their respective control (ctrl) neurons 19–2; actin-beta (ACTB) was used as a loading control and the relative intensity of each band is indicated below the blots; pF, picofarad; MΩ, megaohm; Hz, hertz; pA, picoampere.

*Figure 6 continued on next page*

*Figure 6 continued*

DOI: https://doi.org/10.7554/eLife.40092.015

The following source data and figure supplements are available for figure 6:

**Source data 1.** Multielectrode array values for familial EHMT2 lines.

DOI: https://doi.org/10.7554/eLife.40092.019

**Source data 2.** Patch-clamp recording values for familial EHMT2 lines.

DOI: https://doi.org/10.7554/eLife.40092.020

**Figure supplement 1.** Yeast complementation assay to estimate the pathogenicity of the missense mutation E78K in the human gene UBE2I.

DOI: https://doi.org/10.7554/eLife.40092.016

**Figure supplement 2.** Electrophysiology of the isogenic pair 19–2 and 19–2-EHMT2$^{StopTag/+}$.

DOI: https://doi.org/10.7554/eLife.40092.017

**Figure supplement 2—source data 1.** Patch-clamp recording values for isogenic EHMT2 lines.

DOI: https://doi.org/10.7554/eLife.40092.018

Interestingly, evaluation of EHMT2 protein abundance revealed a clear decrease in the mutant lines 38B and 38E, as compared to the control 37E (*Figure 6C*). This suggests that a reduced expression of *EHMT2* increases spontaneous spiking activity and sEPSC frequency of glutamatergic neurons.

## EHMT2$^{-/+}$ CRISPR-isogenic pair confirms neuronal hyperactivity

Since the prediction of damage extent of the frameshift variant K1164Nfs on the function of EHMT2 may not be accurate, we used our StopTag insertion strategy in iPSC line 19–2, and targeted exon 20 of the transcript ENST00000375537.8 of *EHMT2* in order to disrupt its expression. In this new isogenic line, wMFR and network burst frequency were also increased in iPSC-derived 19–2-*EHMT2*$^{StopTag/+}$ neurons compared to control 19–2, around week 10 PNI and beyond (*Figure 6D* and *Figure 6—figure supplement 2A*). This increased activity in mutant neurons occurred later than that observed in the familial lines 38B/E, possibly due to the more active 19–2 line. Accordingly, EHMT2 protein levels were reduced by half in mutant cells (*Figure 6E*). We also performed patch-clamp recordings on these neurons at day 21–25 PNI, as above. We did not detect any significant change in sEPSC frequency and amplitude at this earlier timepoint, similar to the MEA experiment. However, intrinsic properties showed a significant increase in capacitance and decrease in input resistance in mutant cells (*Figure 6—figure supplement 2B*). These observations suggest that the mutant neurons at 3–4 weeks PNI potentially have a faster maturation rate, however, this phenotype is most pronounced in the hyperactivity recorded by MEAs later at 9–11 weeks PNI. These results support the conclusion that the inactivation of one allele of *EHMT2* significantly increases spontaneous network activity of excitatory neurons, with possible effects on the neuronal maturation process.

## Discussion

In order to establish a scalable iPSC-derived neuron paradigm to study ASD, we selected 12 well-characterized families bearing assumed etiologic variants in ASD-relevant genes, and CNV loci. Per family, we established one to four different fully-characterized and normal iPSC lines from typically one individual with ASD, and one unaffected (non-ASD) sex-matched member. Simultaneous multi-line electrophysiological evaluation revealed hyperactivity of the simple-variant *CNTN5*$^{-/+}$ iPSC-derived glutamatergic neurons in two independent genetic backgrounds. Moreover, isogenic-MEA and patch-clamp recordings confirmed synaptic hyperactivity of iPSC-derived neurons with disruptive mutations in *EHMT2,* also in two different genetic backgrounds.

To increase the modeling scalability of complex genetic disorders such as ASD while optimizing statistical power, several parameters require careful consideration. Given substantial variation in reprogramming and neuronal differentiation efficiencies, sample size is important to control. It was recently proposed that inter-individual variation, that is the number of probands with similar genetic variants, is more important to consider than intra-individual variation, that is the number of iPSC clones derived from the same individual (*Hoffman et al., 2019*). Aiming at multi-variant phenotyping, we tested one or two probands per deficient gene, however, we were able to create an isogenic pair in a different genetic background for the two highly relevant genes, that is, *CNTN5*

(*Lionel et al., 2011*; *Mercati et al., 2017*; *van Daalen et al., 2011*) and *EHMT2* (*Deimling et al., 2017*; *Kleefstra et al., 2005*; *Zylicz et al., 2015*), thereby controlling inter-individual variation. We derived two independent iPSC clones per participants to regulate intra-individual variation.

Another important parameter to consider is the cellular homogeneity of neuronal cultures. We preferred to use the NGN2 system over classic dual-SMAD inhibition protocols because in our experience it represents an advantage in terms of cellular homogeneity. It is also faster than other protocols and produces much higher proportion of glutamatergic neurons that can be studied for ASD (*Canitano and Pallagrosi, 2017*; *Habela et al., 2016*) or other neurological disorders (*Lin et al., 2018*). We assume that most NGN2-neurons are glutamatergic based on the data presented in the original publication establishing this technique (*Zhang et al., 2013*), and on our previous publication using high-cell density RNAseq assessment (*Deneault et al., 2018*). Moreover, we have treated several of our cultures at the end of the MEA experimentation with different neurotoxins (CNQX, PTX, TTX) to show that most neurons are glutamatergic and not GABAergic, for different lines. In addition, our patch-clamp recordings have demonstrated that these neurons exhibit the properties of excitatory neurons.

Characterization of neuronal composition and survival when MEA is performed is difficult to achieve with high accuracy. Our strategy involved using several technical (3 to 12 per independent experiment) and biological (up to 4) replicates to best compensate for inter-well and inter-iPSC neuronal induction variations (*Supplementary File 3*). It is possible that some phenotypes were missed, for example in our families without MEA phenotypes, since we cannot exclude the possibility that differences in cell number or composition across individuals in a family actually masked potential MEA phenotypes. MEA phenotypes may not always predict electrophysiological deficits and vice versa, as evidenced for *ASTN2* in our recent publication (*Deneault et al., 2018*). Since the familial controls are often less active, this screen might be biased towards the identification of hyperactive phenotypes rather than hypoactive. However, we have previously detected hypoactive phenotypes in isogenic KO line 19–2 at week 8 and before (*Deneault et al., 2018*). Line 19–2 is generally more active than most other familial lines, and this may have delayed emergence of the hyperactive phenotype in isogenic cells until week 10. We suggest a developmental time course covering several timepoints for each family moving forward using MEA. For specific lines, different timepoints may be sufficient.

An increased neuronal activity, for example MFR in mutant lines 38B/E, might indicate alterations in synaptic function and/or maturation. We have presented the MFR for all tested cell lines in *Figure 3—figure supplement 1*, in support of the wMFR in *Figure 3*. The wMFR is defined as the MFR divided by the number of active electrodes per well. If there is significant failure of electrode activation in a well, for example due to differences in cell survival, dispersion or adhesion, those data are excluded. For example, the increased MFR observed in $EHMT2^{-/+}$ lines could be due to a better capacity to survive, disperse or adhere than $EHMT2^{+/+}$ cells, without affecting synaptic activity. However, excluding all inactive electrodes would then result in a comparable wMFR between mutant and control cells, which was not the case. Indeed, both MFR and wMFR were significantly higher in $EHMT2^{-/+}$ cells. That does not exclude the possibility of a better survival, dispersion or adhesion, but it is likely not the only reason for the observed increase in spiking activity, suggesting greater synaptic activity, as supported by patch-clamp recordings. Indeed, we used patch-clamp recordings to show that sEPSC frequency was significantly increased in mutant line 38E compared to control line 37E. We believe these results directly support synaptic alteration as one of the possible causes for the increased neuronal activity measured by MEA. However, a detailed analysis of cell maturation will be required for each different cell line involved in this study to clarify this issue. It will be interesting in the future to investigate the possible mechanisms involved in the decreased activity observed in *THRA*-mutant neurons (*Figure 3D*), after validation by patch-clamp recordings.

*ANOS1* (Anosmin 1) is a glycoprotein of the extracellular matrix including four consecutive fibronectin type III domains. Loss-of-function variants in *ANOS1* were shown to cause the Kallmann syndrome, which is characterized by congenital hypogonadotropic hypogonadism associated with anosmia, delayed puberty and infertility (*Dodé and Hardelin, 2009*). Defects in the migration of gonadotropin-releasing hormone (GnRH) neurons were observed during embryonic development, as well as morphological changes in the basal forebrain cortex (*Manara et al., 2014*). In human, a proband carrying the nonsense variant R423X in *ANOS1,* and presenting clinical hypogonadotropic hypogonadism, was also diagnosed with ASD (*Jiang et al., 2013*), suggesting a link between

*ANOS1* and ASD. Despite the absence of significant MEA results at late recordings, neuronal membrane current defects were validated using patch-clamp recordings in an isogenic pair (*Figure 5D*). These results indicate that glutamatergic neuron activity is also influenced by *ANOS1*, which represents a risk gene for ASD.

*CNTN5* (Contactin 5) is an immunoglobulin cell adhesion molecule, with four fibronectin type III domains, involved in neurite outgrowth and axon connection in cortical neurons, and was associated with ASD (*van Daalen et al., 2011*). Different CNVs affecting *CNTN5* have been associated with ASD and ADHD, with increased occurrence of hyperacusis (*Lionel et al., 2011*; *Mercati et al., 2017*). The molecular mechanisms through which heterozygous loss of *CNTN5* increases neuronal activity in vitro (*Figure 4A–B*) remains to be elucidated. Gene editing of the 676 kb deletion, as found in lines 27H and 27N (*Figure 1D* and *Table 1*), to obtain isogenic controls may be challenging due to the size, but this approach might eventually be applied.

Using a yeast complementation assay (*Figure 6—figure supplement 1*), we estimated that the de novo missense variant E78K in UBE3I was not responsible for the electrophysiological phenotypes observed in participant 6-0393-003 (*Figure 6A–B*). We were then prompted to investigate further the potential role of the frameshift variant K1164Nfs in *EHMT2*. EHMT2 (G9a) is a histone methyltransferase (HMTase) that forms a complex with EHMT1 (GLP) to catalyze mono- and dimethylation of lysine nine on histone H3 (H3K9me1/2) (*Rice et al., 2003*). Of note, EHMT1 protein sequence is highly similar to EHMT2 (*Deimling et al., 2017*). Actually, EHMT1 haploinsufficiency is involved in intellectual disability (ID) and ASD as part of the Kleefstra syndrome (*Kleefstra et al., 2005*). EHMT2 represses pluripotency genes in embryonic stem cells (*Zylicz et al., 2015*) and potentially acts as both repressor of neural progenitor genes and activator of neuronal differentiation (*Deimling et al., 2017*). The impact of the single base deletion in *EHMT2* (K1164Nfs) on the protein function remains to be determined (see *Table 1* for details). The frameshift is computationally predicted to extend the protein rather than truncating it, by utilizing sequence in the 3'UTR. However, it is located exactly at the beginning of the post-SET domain, that is at position 1164 of EHMT2. The resulting change in the downstream protein sequence completely disrupts three conserved cysteine residues in the post-SET domain that normally form a zinc-binding site with a fourth conserved cysteine close to the SET domain (*Zhang et al., 2003*). Since these three conserved cysteine residues are essential for HMTase activity, as replacement with serine abolished HMTase activity (*Zhang et al., 2002*), we suspect that this HMTase activity of EHMT2 is defective in our mutant glutamatergic neurons and potentially related to the observed hyperactivity (*Figure 6*). Upon our further validation experiment using a CRISPR-derived isogenic system and an unrelated genetic background (*Figure 6D*), we propose that *EHMT2* impacts the synaptic function of glutamatergic neurons through H3K9me1/2 catalyzing ability. Further experiments might clarify this possibility, such as CRISPR-correction of the K1164Nfs point mutation in lines 38B and 38E to obtain isogenic controls.

Overall, this study highlights a way to improve the scalability of testing multiple iPSC-derived neuronal lines with various ASD-risk variants. Furthermore, our work demonstrates that for future studies to capture and characterize the electrophysiological impact of ASD variants on human iPSC-NGN2 neurons, it is most beneficial to include both MEA and patch-clamp experiments, across multiple timepoints. Analyzing multiple mutations and genes at once can lead to the identification of potential endophenotypes, in this case neuronal hyperactivity. This work revealed that inactivation of at least one allele of *CNTN5* or *EHMT2* significantly intensifies excitatory neuron synaptic activity in vitro. Such phenotype offers the possibility to implement NGN2-based high-throughput drug screening strategies (*Cheng et al., 2017*) combining MEA (*Tukker et al., 2018*) and lines 38B/38E for instance, to discover molecules that may compensate for neuronal hyperactivity.

## Materials and methods

### Ethics for human experiments

Under the approval of the Canadian Institutes of Health Research Stem Cell Oversight Committee and the Research Ethics Board (REB) at the Hospital for Sick Children, Toronto, Canada, iPSCs were generated from dermal fibroblasts or CD34 +blood cells. Three different informed consent forms for iPSC derivation and publication were obtained: i) Research Consent Form for Parent/Legal Guardians (of an individual with a neurologic condition); ii) Research Consent Form for Unaffected

Individuals; iii) Assent form (for individuals with a neurologic condition). These consent forms describe in details the purpose of the research, the description of the research, the potential harms, the potential benefits, confidentiality, storage of the research samples, participation, reimbursement, sponsorship, and declaration of conflict of interest; REB approval file 1000012015.

## Skin fibroblasts culture

Skin-punch biopsies were obtained from the upper back area by a clinician at The Hospital for Sick Children. Samples were immersed in 14 ml of ice-cold Alpha-MEM (Wisent Bioproducts) supplemented with penicillin 100 Units/ml and streptomycin 100 µg/ml (ThermoFisher), and transferred immediately to the laboratory at The Centre for Applied Genomics (TCAG). Each biopsy was cut into ~1 mm$^3$ pieces with disposable scalpel in a 60 mm dish. 5 ml of collagenase 1 mg/ml (Sigma, Canada) was added and the dish was placed in 37°C incubator for 1:45 hr. Skin pieces and collagenase were then transferred to a 15 ml tube, and centrifuged at 300 g for 10 min. Supernatant was removed, 5 ml of trypsin 0.05%/EDTA 0.53 mM (Wisent Bioproducts) was added, and the mix was pipetted up and down several times to break up tissue and placed in 37°C incubator for 30 min. After incubation, the mix was centrifuged at 300 g for 10 min, and supernatant was removed leaving 1 ml. The pellet was pipetted up and down vigorously to break to the pieces without creating bubbles. The mix was transferred in a T-12.5 flask along with 5 ml of Alpha-MEM, 15% Fetal Bovine Serum (FBS; Wisent Bioproducts), penicillin 100 Units/ml and streptomycin 100 µg/ml (ThermoFisher), and placed in 37°C incubator for about a week until 100% confluence. Cultured cells were fed every 5–7 days if not confluent. Once confluent, cells were passed into three 100 mm dishes to expand, and frozen in liquid nitrogen.

## Reprogramming fibroblasts using integrative virus

Reprogramming of skin fibroblasts was performed using retroviral and lentiviral vectors. Retroviral vectors encoding *POU5F1*, *SOX2*, *KLF4*, *MYC*, and lentiviral vectors encoding the pluripotency reporter EOS-GFP/Puro$^R$ were used and obtained as described (*Hotta et al., 2009*).

## Reprogramming fibroblasts using non-integrative Sendai virus

Reprogramming of fibroblasts via Sendai virus was performed at the Centre for Commercialization of Regenerative Medicine (CCRM) using CytoTune-iPS 2.0 Sendai Reprogramming Kit (Thermo-Fisher). Fibroblasts were cultured in fibroblast expansion media (Advanced DMEM; 10% FBS; 1X L-Glutamine; 1X pen/strep – Thermo Fisher). The desired number of wells for reprogramming from a 24-well plate was coated with 0.1% gelatin. Fibroblasts were dissociated using Trypsin (Thermo-Fisher) and allowed to settle overnight. Virus multiplicity of infection (MOI) was calculated and viruses combined according to number of cells available for reprogramming and manufacturer's protocol. 24 hr after transduction, media was changed to wash away viruses. Media was additionally changed on day 3 and 5 after transduction. 6 days after transduction, 6-well plates were coated with Matrigel(Corning). Cells were removed from the 24-well plate using Accutase (ThermoFisher) and plated on Matrigel in expansion media. 24 hr later, media was replaced with E7 media (StemCell-Technologies). Cells were monitored and fed daily with E7. Once colonies were of an adequate size and morphology to pick, individual colonies were picked and plated into E8 media (StemCellTechnologies). Clones growing well were further expanded and characterized using standard assays for pluripotency, karyotyping, genotyping and mycoplasma testing. Directed differentiation was performed using kits for definitive endoderm, neural and cardiac lineages (all ThermoFisher).

## Peripheral blood mononuclear sells (PBMCs) isolation from peripheral blood and enrichment of CD34 +cells

Whole peripheral blood was processed at CCRM using Lymphoprep (StemCellTechnologies) in a SepMate tube (StemCellTechnologies) according to manufacturer's instructions. The sample was centrifuged (10 min at 1200 g). The top layer containing PBMCs was collected and mixed with 10 mL of the PBS/FBS mixture and centrifuged (8 min at 300 g). The PBMC's collected at the bottom of the tube were washed, counted and resuspended in PBS/FBS mixture. CD34 +cells were then isolated using the Human Whole Blood/Buffy Coat CD34 +Selection kit according to manufacturer's instructions (StemCellTechnologies). Isolated cells were expanded in StemSpan SFEM II media

(StemCellTechnologies) and StemSpan CD34 +Expansion Supplements (StemCellTechnologies) prior to reprogramming.

## Reprogramming PBMC using non-Integrative Sendai virus

Reprogramming of CD34 +PBMCs was performed at CCRM using CytoTune-iPS 2.0 Sendai Reprogramming Kit. Expanded cells were spun down and resuspended in StemSpan SFEM II media and StemSpan CD34 +Expansion Supplements, and placed in a single well of a 24-well dish. Virus MOI was calculated and viruses combined according to number of cells available for reprogramming and manufacturer's protocol. The virus mixture was added to cells, and washed off 24 hr after infection. 48 hr after viral delivery, cells were plated in 6-well plates in SFII and transitioned to ReproTESR for the duration of reprogramming. Once colonies were of an adequate size and morphology to pick, individual colonies were picked and plated into E8. Clones growing well were further expanded and characterized as explained above.

## iPSC maintenance

All iPSC lines were maintained on matrigel (Corning) coating, with complete media change every day in mTeSR (StemCellTechnologies). ReLeSR (StemCellTechnologies) was used for passaging. Accutase (InnovativeCellTechnologies) and 10 µM Rho-associated kinase (ROCK) inhibitor (Y-27632; StemCellTechnologies) were used for single-cell dissociation purposes.

## Gene editing

For point mutation correction in 18C line, we used the type II CRISPR/Cas9 double-nicking (Cas9D10A) system with two guide RNA (gRNAs) to reduce off-target activity. We devised the gRNA sequences using tools available at http://crispr.mit.edu/. We designed a HDR-based method using a synthesized single-stranded oligonucleotide (ssODN) template to replace the point mutation with the reference nucleotide. To prevent damage to the correct sequence, a silent mutation was introduced in the ssODN close to the proto-adjacent motif (PAM) of the reverse gRNA (gRNA-), which commands Cas9D10A to nick the plus strand, given that ssODN was synthesized as plus strand. All the CRISPR machinery was introduced into iPSC by nucleofection. Screening for correction of the appropriate base pair was based on absolute quantification of allele frequency using droplet digital PCR (ddPCR). Enrichment of corrected cells was obtained through sib-selection step cultures in 96-well plate format, as adapted from (*Miyaoka et al., 2014*), until a well containing 100% of corrected alleles was identified. For insertion of premature stop codon in 19–2 cells, ribonucleoprotein (RNP) complex was used as a vector to deliver the CRISPR machinery, along with one sgRNA and Cas9 nuclease, for each target gene. Design of sgRNA and ssODN for HDR, nucleofection and isolation of edited lines were described (*Deneault et al., 2018*).

## Lentivirus production

$7.5 \times 10^6$ HEK293T cells were seeded in a T-75 flask, grown in 10% fetal bovine serum in DMEM (Gibco). The next day, cells were transfected using Lipofectamine 2000 with plasmids for gag-pol (10 µg), rev (10 µg), VSV-G (5 µg), and the target constructs FUW-TetO-Ng2-P2A-EGFP-T2A-puromycin or FUW-rtTA (15 µg; gift from T.C. Südhof laboratory) (*Zhang et al., 2013*). Next day, the media was changed. The day after that, the media was spun down in a high-speed centrifuge at 30,000 g at 4°C for 2 hr. The supernatant was discarded and 50 µl PBS was added to the pellet and left overnight at 4°C. The next day, the solution was triturated, aliquoted and frozen at −80°C.

## Differentiation into glutamatergic neurons

$5 \times 10^5$ iPSCs/well were seeded in a matrigel-coated 6-well plate in 2 ml of mTeSR supplemented with 10 µM Y-27632. Next day, media in each well was replaced with 2 ml fresh media plus 10 µM Y-27632, 0.8 µg/ml polybrene (Sigma), and the minimal amount of NGN2 and rtTA lentiviruses necessary to generate 100% GFP +cells upon doxycycline induction, depending on prior titration of a given virus batch. The day after, virus-containing media were replaced with fresh mTeSR, and cells were expanded until near-confluency. Newly generated 'NGN2-iPSCs' were detached using accutase, and seeded in a new matrigel-coated 6-well plate at a density of $5 \times 10^5$ cells per well in 2 ml of mTeSR supplemented with 10 µM Y-27632 (day 0 of differentiation). Next day (day 1), media in

each well was changed for 2 ml of CM1 [DMEM-F12 (Gibco), 1x N2 (Gibco), 1x NEAA (Gibco), 1x pen/strep (Gibco), laminin (1 μg/ml; Sigma), BDNF (10 ng/μl; Peprotech) and GDNF (10 ng/μl; Peprotech) supplemented with fresh doxycycline hyclate (2 μg/ml; Sigma) and 10 μM Y-27632. The day after (day 2), media was replaced with 2 ml of CM2 [Neurobasal media (Gibco), 1x B27 (Gibco), 1x glutamax (Gibco), 1x pen/strep, laminin (1 μg/ml), BDNF (10 ng/μl) and GDNF (10 ng/μl)] supplemented with fresh doxycycline hyclate (2 μg/ml) and puromycin (5 μg/ml for 19–2-derived cells, and 2 μg/ml for 50B-derived cells; Sigma). Media was replaced with CM2 supplemented with fresh doxycycline hyclate (2 μg/ml). The same media change was repeated at day 4. At day 6, media was replaced with CM2 supplemented with fresh doxycycline hyclate (2 μg/ml) and araC (10 μM; Sigma). Two days later, these day eight post-NGN2-induction (PNI) neurons were detached using accutase and ready to seed for subsequent experiments, as described below.

## Multi-electrode array (MEA)

48-well opaque-bottom MEA plates (Axion Biosystems, M768-KAP-48), 16 electrodes per well, were coated with filter-sterilized 0.1% polyethyleneimine solution in borate buffer pH 8.4 for 1 hr at room temperature, washed four times with water, and dried overnight. 120,000 'day8-dox' neurons/well were seeded in a 5 ul drop of CM2 media at the centre of each well, then covered with 250 μl CM2 media after one hour in the incubator. The day after, 5,000 mouse astrocytes/well were seeded on top of neurons in 50 μl/well CM2 media. Astrocytes were prepared from postnatal day 1 CD-1 mice as described (*Kim and Magrané, 2011*). Media was half-changed once a week with CM2 media. Every week post-seeding, the electrical activity of the MEA plates was recorded using the Axion Maestro MEA reader (Axion Biosystems). The heater control was set to warm up the reader at 37°C. Each plate was first incubated for 5 min on the pre-warmed reader, then real-time spontaneous neural activity was recorded for 5 min using AxIS 2.0 software (Axion Biosystems). A bandpass filter from 200 Hz to 3 kHz was applied. Spikes were detected using a threshold of 6 times the standard deviation of noise signal on electrodes.

Offline advanced metrics were re-recorded and analysed using Axion Biosystems Neural Metric Tool. An electrode was considered active if at least five spikes were detected per minute. Single electrode bursts were identified as a minimum of five spikes with a maximum interspike interval (ISI) of 100 milliseconds. Network bursts were identified as a minimum of 10 spikes with a maximum ISI of 100 milliseconds covered by at least 25% of electrodes in each well. No non-active well was excluded in the analysis. After the last reading, each well was treated with three synaptic antagonists: GABA$_A$ receptor antagonist picrotoxin (PTX; Sigma) at 100 μM, AMPA receptor antagonist 6-cyano-7-nitroquinoxaline-2,3-dion (CNQX; Sigma) at 60 μM, and sodium ion channel antagonist tetrodotoxin (TTX; Alomone labs) at 1 μM. The plates were recorded consecutively, 5–10 min after addition of the antagonists. A 60 min recovery period was allowed in the incubator at 37°C between each antagonist treatment and plate recording.

## Patch-clamp recordings

Day 3 PNI neurons were replated at a density of 100,000/well of a poly-ornithin/laminin coated coverslips in a 24-well plate with CM2 media. On day 4, 50,000 mouse astrocytes were added to the plates and cultured until day 21–28 PNI for recording. At day 10, CM2 was supplemented with 2.5% FBS in accordance with (*Zhang et al., 2013*). Whole-cell recordings (BX51WI; Olympus) were performed at room temperature using an Axoclamp 700B amplifier (Molecular Devices) from borosilicate patch electrodes (P-97 puller; Sutter Instruments) containing a potassium-based intracellular solution (in mM): 123 K-gluconate, 10 KCL, 10 HEPES; 1 EGTA, 2 MgCl$_2$, 0.1 CaCl$_2$, 1 Mg-ATP, and 0.2 Na$_4$GTP (pH 7.2). 0.06% sulpharhodamine dye was added to select neurons for visual confirmation of multipolar neurons. Composition of extracellular solution was (in mM): 140 NaCl, 2.5 KCl, 1 1.25 NaH$_2$PO$_4$, 1 MgCl$_2$, 10 glucose, and 2 CaCl$_2$ (pH 7.4). Whole cell recordings were clamped at −70 mV using Clampex 10.6 (Molecular Devices), corrected for a calculated −10 mV junction potential and analyzed using the Template Search function from Clampfit 10.6 (Molecular Devices). Following initial breakthrough and current stabilization in voltage clamp, the cell was switched to current clamp to monitor initial spiking activity and record the membrane potential (cc = 0,~1 min post-breakthrough). Bias current was applied to bring the cell to ~70 mV whereby increasing 5 pA current steps were applied (starting at −20 pA) to generate the whole cell resistance and to elicit action

potentials. Data were digitized at 10 kHz and low-pass filtered at 2 kHz. Inward and outward currents were recorded in whole-cell voltage clamp in response to consecutive 10 mV steps from −90 mV to +40 mV.

### Yeast complementation assay

The method for the yeast complementation assay was described previously (*Sun et al., 2016*).

### Antibodies and western blotting

Cells were washed in ice-cold PBS and total protein was extracted in RIPA supplemented with proteinase inhibitor cocktail, and homogenized. Equivalent protein mass was loaded on gradient SDS-PAGE (4–12%) and transferred to Nitrocellulose membrane Hybond ECL (GE HealthCare). Primary antibodies used were rabbit anti-CNTN5 (Novus, NBP1-83243) and rabbit anti-EHMT2/G9A (Abcam, ab185050). HRP-conjugated secondary antibodies (Invitrogen) were used and the membranes were developed with SuperSignal West Pico Chemiluminescent Substrate (Pierce). Images acquired using ChemiDoc MP (BioRad) and quantified using software Imagelab v4.1 (BioRad). Western Blots were repeated at least twice for each biological replicate.

### Mycoplasma testing

All cell lines were regularly tested for presence of mycoplasma using a standard method (*Otto et al., 1996*).

## Acknowledgements

The authors wish to acknowledge the resources of MSSNG (www.mss.ng), Autism Speaks and The Centre for Applied Genomics at The Hospital for Sick Children, Toronto, Canada. We also thank the participating families for their time and contributions to this database, as well as the generosity of the donors who supported this program. We also thank Drs. Melissa Carter, Wendy Roberts, Brian Chung and Rosanna Weksberg for obtaining skin biopsies and blood work, and the families for volunteering. We also thank Tara Paton, Guillermo Casallo, Barbara Kellam, Ny Hoang and Sylvia Lamoureux for technical help; TC Südhof for the NGN2/rtTA lentiviral constructs.

## Additional information

#### Competing interests

Stephen W Scherer: Serves on the Scientific Advisory Committees of Population Bio and Deep Genomics, and intellectual property originating from his research and held at the Hospital for Sick Children is licensed to Lineagen, and separately Athena Diagnostics. The other authors declare that no competing interests exist.

#### Funding

| Funder | Author |
| --- | --- |
| Canadian Institutes of Health Research | Fritz Roth<br>Karun K Singh<br>James Ellis<br>Stephen W Scherer |
| Canadian Institute for Advanced Research | Fritz Roth<br>Stephen W Scherer |
| Canada Foundation for Innovation | Fritz Roth<br>James Ellis<br>Stephen W Scherer |
| National Institutes of Health | Fritz Roth<br>James Ellis<br>Stephen W Scherer |

| Ontario Brain Institute | Karun K Singh<br>James Ellis<br>Stephen W Scherer |
| Natural Sciences and Engineering Research Council of Canada | Karun K Singh |
| Province of Ontario Neurodevelopmental Disorders | Karun K Singh<br>James Ellis |
| Ontario Research Fund | James Ellis<br>Stephen W Scherer |
| Genome Canada | Stephen W Scherer |
| University of Toronto McLaughlin Centre | Stephen W Scherer |
| Autism Speaks | Stephen W Scherer |
| Hospital for Sick Children | Stephen W Scherer |

The funders had no role in study design, data collection and interpretation, or the decision to submit the work for publication.

## Author contributions

Eric Deneault, Conceptualization, Data curation, Formal analysis, Validation, Investigation, Visualization, Methodology, Writing—original draft, Writing—review and editing; Muhammad Faheem, Formal analysis, Validation, Investigation, Methodology; Sean H White, Data curation, Formal analysis, Validation, Investigation, Methodology, Writing—review and editing; Deivid C Rodrigues, Formal analysis, Validation, Methodology, Writing—review and editing; Song Sun, Data curation, Formal analysis, Validation, Methodology, Writing—review and editing; Wei Wei, Alina Piekna, Tadeo Thompson, Validation, Investigation, Methodology; Jennifer L Howe, Data curation, Formal analysis, Validation, Investigation, Project administration, Writing—review and editing; Leon Chalil, Vickie Kwan, Data curation, Formal analysis, Validation, Methodology; Susan Walker, Conceptualization, Data curation, Formal analysis, Validation, Investigation, Project administration, Writing—review and editing; Peter Pasceri, Validation, Methodology, Project administration; Frederick P Roth, Resources, Data curation, Supervision, Funding acquisition, Validation, Investigation, Writing—review and editing; Ryan KC Yuen, Conceptualization, Data curation, Formal analysis, Validation, Investigation, Methodology, Writing—review and editing; Karun K Singh, James Ellis, Stephen W Scherer, Conceptualization, Resources, Supervision, Funding acquisition, Investigation, Visualization, Writing—original draft, Writing—review and editing

## Author ORCIDs

Eric Deneault (iD) http://orcid.org/0000-0002-6875-0380
Karun K Singh (iD) https://orcid.org/0000-0001-8301-3533
James Ellis (iD) http://orcid.org/0000-0002-4400-0091
Stephen W Scherer (iD) http://orcid.org/0000-0002-8326-1999

## Ethics

Human subjects: All ASD and related control-participants were initially consented for WGS and upon return of genetic results, then consented for the iPSC study, using approved protocols through the Research Ethics Board at the Hospital for Sick Children. Under the approval of the Canadian Institutes of Health Research Stem Cell Oversight Committee and the Research Ethics Board (REB) at the Hospital for Sick Children, Toronto, Canada, iPSCs were generated from dermal fibroblasts or CD34 + blood cells. Three different informed consent forms for iPSC derivation and publication were obtained: i) Research Consent Form for Parent/Legal Guardians (of an individual with a neurologic condition); ii) Research Consent Form for Unaffected Individuals; iii) Assent form (for individuals with a neurologic condition). These consent forms describe in details the purpose of the research, the description of the research, the potential harms, the potential benefits, confidentiality, storage of

the research samples, participation, reimbursement, sponsorship, and declaration of conflict of interest; REB approval file 1000012015.

### Decision letter and Author response
Decision letter https://doi.org/10.7554/eLife.40092.026
Author response https://doi.org/10.7554/eLife.40092.027

## Additional files

### Supplementary files
• Supplementary file 1. Genomic coordinates of the genetic variant(s) associated with each participant.
DOI: https://doi.org/10.7554/eLife.40092.021

• Supplementary file 2. Characterization of pluripotency, differentiation potential and karyotype of iPSC lines. n/a, not available; STR, short tandem repeat.
DOI: https://doi.org/10.7554/eLife.40092.022

• Supplementary file 3. Number of different wells per sample for each different MEA plates. *Independent experiments imply independent infections with NGN2 viruses of iPSCs at different passages, entailing completely independent inductions.
DOI: https://doi.org/10.7554/eLife.40092.023

• Transparent reporting form
DOI: https://doi.org/10.7554/eLife.40092.024

### Data availability
All MEA data, iPSC lines, and other data and bio-resources described in the manuscript will be publicly available upon request at the time of publication. Should an appropriate receptor repository for de-identified and protected large-scale MEA data be found, it will also be deposited there. Source summary files of the underlying data used to generate Figures 3, 4, 5 and 6 are also provided with the paper. Requests for additional information, or materials, should be made by email to the last listed senior corresponding author (S.W.S.). Upon confirming these such requests are part of an institutionally-approved research project, the resources will be transferred under a standard Materials Transfer Agreement signed between the sending and receiving institutions.

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
