## [Decision Letter]

[**Editorial note:** This article has been through an editorial process in which the authors decide how to respond to the issues raised during peer review. The Reviewing Editor's assessment is that all the issues have been addressed.]

Thank you for submitting your article "*CNTN5*^-/+^ or *EHMT2*^-/+^ iPSC-derived neurons from individuals with autism develop hyperactive neuronal networks" for consideration by *eLife*. Your article has been reviewed by two peer reviewers, and the evaluation has been overseen by a Reviewing Editor and Huda Zoghbi as the Senior Editor. The reviewers have opted to remain anonymous.

The Reviewing Editor has highlighted the concerns that require revision and/or responses, and we have included the separate reviews below for your consideration. If you have any questions, please do not hesitate to contact us.

Summary:

This report of IPSC derived neurons with autism-risk gene mutations was highly regarded and represents a novel investigation that many groups have sought to achieve. The finding of hyperactive neuronal networks is consistent with current views of the disorder and is a strong contribution to the field.

Major concerns:

The two referees carried out an extensive evaluation of the manuscript and raised multiple questions about the data acquisition and the mechanism of action. It was felt the multi-electrode array (MEA) protocols need more explanation and assurance of reliability. In summary, both reviewers requested the following additions to strengthen the manuscript-

1) Inclusion of existing MEA data across all 12 donors

2) Consideration of the efficiency and yield of NGN2 donors across donors and experiments.

The MEA data should be a full figure. A full data set will be key in understanding line developmental time course, and variability between wells, lines, and individuals. It was also recommended that there is a need to discuss the apparent limitations of their strategy and implementation.

Separate reviews (please respond to each point):

*Reviewer #1:*

In their manuscript entitled "*CNTN5*^-/+^ or *EHMT2*^-/+^ iPSC-Derived Neurons from Individuals with Autism Develop Hyperactive Neuronal Networks," Deneault et al. describe a collection of patient-specific hiPSC lines with heterozygous mutations in 12 different ASD associated loci (15 cases, each with at least one related unaffected relative and/or an CRISPR-edited hiPSC line as control; 27 hiPSC lines total). hiPSCs were NGN2-induced to excitatory neurons and then compared by multielectrode array; key findings for *CNTN5*^-/+^ and *EHMT2*^-/+^ were evaluated by electrophysiology. This collection of ASD and related control derived hiPSCs represents a valuable resource to the community, worthy of describing in *eLife*. The value of such a collection would be enhanced by making publically available genotype data from each donor, all hiPSC lines and all MEA data. While an impressive degree of work is described, no insights into why some patient neurons, but not others, showed activity deficits or the mechanism linking genotype to neuronal function is included. With major revisions, this report will be suitable for publication at *eLife*.

Major detail:

1) ASD neuronal phenotypes – The authors conducted MEA across all 25 hiPSC lines (Supplementary file 3) but do not seem to comprehensively present this data anywhere, focusing instead on *CNTN5*^-/+^ and *EHMT2*^-/+^. The findings (even if null) should be aggregated and clearly presented as a main figure. Additionally, the authors should make all raw MEA data and all hiPSC lines widely available upon publication.

2) Inter-individual and intra-individual variability – The authors have generated perhaps the largest systematic MEA dataset across unrelated controls and ASD patients with different rare mutations. While they aggregate all control data in SI Figure 2 and 3, it would be tremendously valuable to the field if they might plot their results by donor as a main figure. Being able to show how MEA activity changes over time, and between donors, would greatly inform our understanding of inter-individual variation in MEA. Similarly, in those cases where the same line was tested on multiple MEA plates, again, it would be useful to present variability.

3) Inter-well and inter-hiPSC neuronal induction efficiency – It is incredibly difficult to assess and adjust for differences in total cell number and cell type composition in MEA experiments, both of which are expected to impact MEA recordings. What was the authors' strategy? It would be helpful if in the discussion they might inform the readers how best to learn from and extend their analyses.

4) CRISPR-engineering – How did the authors rule out off-target effects following CRISPR-editing to produce isogenic pairs?

5) Phenotypic characterization – Increased neuronal activity might indicate alterations in synaptic function and/or maturation. For the two genes focused on within the manuscript, *CNTN5*^-/+^ or *EHMT2*^-/+^, the authors should provide attempt to resolve this question (even a time course of transcriptomic changes over 6-weeks of NGN2-induction).

6) Mechanism – Can the authors speculate as to the mechanism through which *CNTN5*^-/+^ or *EHMT2*^-/+^ increase neuronal activity? Possibly by linking their results to existing animal models? Perhaps RNAseq analyses might provide insights into the downstream changes in patient-derive neurons?

7) Data sharing – The authors have generated a great resource that is likely to be of great interest to the community. The value of such a collection would be enhanced by making publically available the inclusion of SNP and exome genotype data from each donor and clarification of the specific hiPSC lines available (at which repository). How many validated hiPSCs per donor are being made available? Through which repository? Will the MEA data be made available?

Minor detail:

1) Clarification of terminology – NGN2-overexpression is best described as "induction" not "differentiation." The authors are cautioned against describing the neurons as "cortical" without clear evidence of patterning.

2) Can the authors clarify in the main text (not just in the Materials and methods) that astrocyte co-culture was used with NGN2-neurons to improve maturation for MEA and electrophysiological.

Additional data files and statistical comments:

Please make available all MEA raw data to the community.

*Reviewer #2:*

This study uses important new tools, specifically human iPSC-derived neurons in models of developing neurons in culture, to define mechanisms that may underlie ASD pathogenesis. Further, they correctly state that: "Proband-specific iPSC-derived neuronal cells indeed provide a useful model to study disease pathology, and response to drugs, but throughput (both iPSC-derived neurons and phenotyping) is low, with costs still high. As a result, so far, only a few iPSC-derived neuronal lines are typically tested in a single study." This indeed has been a limitation, so that evidence of new technologies and approaches would be a welcome advance to speed development of the field. Furthermore, they correctly point out that iPSC-derived neurons carry "the precise repertoire of rare and common genetic variants as the donor proband,".… and "represent the best genetic mimic of proband neurons for functional and mechanistic studies." This approach does overcome many limitations of more narrow, experimentally designed lines where only known genetic factors can be specifically manipulated. So the field needs to move in this direction.

The authors have performed a tremendous amount of experimental work on numerous genetically altered human iPSC-derived neuronal lines, and these lines remain as a resource for the broader community, which may benefit. The large numbers of lines and experiments provide some level of reproducibility and do identify some very specific abnormalities in neuronal function. However there are a number of deficiencies in the manuscript that diminish enthusiasm. There are major concerns regarding inadequate descriptions of methods and data analysis, making it difficult for another investigator to reproduce this approach as presented. Without this information, it is difficult to assess the utility of the recommended multi-electrode array (MEA) protocols and analytic techniques, since the MEA was informative for only 3 of the 12 families. It is not clear that this low success rate represents an absence of circuit abnormalities in the other 9 families, or alternatively, a technical and analytic design that does not distinguish changes in circuit activity, from abnormalities in cell survival, cell aggregation, cell distribution, electrode contact, or cell type composition. The absence of any cellular analysis that underlies the MEA outcomes seems to place this work in the domain of preliminary studies. Specific comments follow:

Abstract: It seems to this reviewer that the Abstract communicates a more positive result than was actually achieved, as regards the "standardized set of procedures," that "used a large-scale multi-electrode array (MEA).…" These statements seem to suggest the value of this approach to the general field. Yet, of the 12 families with genetic variants, only 3 of the 12 MEA studies revealed any differences, which does not necessarily recommend this as a mainstay approach. The alternative approach, using single cell electrophysiology, defined only one additional pathology. The Abstract should be modified to change the emphasis, and consider stating the success rate of 33%.

Materials and methods:

The authors do not mention performing a power analysis in designing the study, and power analysis is not discussed in the statistical methods. There is concern that the inherent variability in the MEA data is not well appreciated nor well controlled, and without this analysis, it cannot be transparently reported.

There is no peer-reviewed, published characterization by the authors of their results of using NEUROG2 expression to induce glutamatergic neurons, characterizations that other cited authors have performed. While the authors state that; "Importantly, we determined that this strategy offers highly uniform differentiation levels between cell lines derived from different participants (Deneault et al., 2018);" that reference is to a manuscript submitted to bioRxiv that has not undergone peer review, and for the current article to be published, analysis of the NEUROG2 induction procedure needs to be presented. The statement that "differentiation levels" were uniform between cell lines, is quite a claim, and it is not at all clear what this means. For example what endpoints were assessed among multiple lines to indicate uniformity? One can imagine, for example, that failure to observe a specific differentiation feature, could reflect that the marker is not expressed by a cell, versus, that it is expressed but at a time point not examined. Such an outcome could reflect changes in i) cell fate, ii) induction of differentiation, iii) survival of a specific cell class, or simply, iv) a delay in its expression in suboptimal culture conditions. There is no characterization of the surviving cell population when MEA analysis of the cultures is performed, so that specific experimental outcomes cannot be related to the cell population under study. Even though the culture bottoms are opaque, immunocytochemistry using fluorescent labeling would allow visualization of cell numbers (DAPI) and cell type (NeuN, MAP2, vGlut, vGAD, etc) by use of the epifluorescence microscope to understand how cell composition at the time of MEA recording impacted the outcomes. This seems a reasonable request since the majority of MEA studies did not reveal group genetic differences, with only 3 of the 12 families demonstrating changes in firing rate and network burst frquencies.

The MEA methods are incomplete as reported, and extensive review of Supplementary file 3 still leaves uncertainties for this reviewer. The authors employ MEA as the single major measure of outcome, but they do not reference their previous use of this technology. It is not adequate to cite other laboratories. The authors should describe more fully the nature of the MEA technology, how the cultures appear in them, the data that is generated, and how it was analyzed. To start, the authors indicate that 48-well opaque-bottom MEA plates (Axion Biosystems) were used. But going to the website there are four (4) "Classic" varieties of this plate, as well as many other formats. Please provide exact catalogue number or numbers. How many electrodes are there in each well? It may be helpful to show a layout/picture for the electrode array, and also pictures of fixed and stained neuronal cultures to define the variability in cell contact to multiple electrodes, and perhaps the extremes of electrode contact they observe. Doing a brief PubMed search, I found two recent 2018 publications that provide images that show large variability of cell growth around fixed electrode arrays (DeRosa et al., Sci Rep 2018; Odawara et al., Sci Rep 2018; also see Hammond et al., BMC Neurosci, 2013). Please state the range of individual MEA wells used in these types of studies: Supplementary file 3 seems to indicate a range from 3 wells to as many as 12. How many independent MEA assay experiments are performed for each line? Supplementary file 3 is confusing, specifically the column labeled "Total nb independent plates." This number coincides with the reported number of different MEA plates (MEA plate IDs are 1-37, with only 21 involved here). Is an independent plate a separate experiment? Usually in iPSC studies, within a single clone (as reported here), independent experiments imply different passages of the same clone, analyzed in subsequent weeks, or after thawing an earlier passage and newly plated. But after NEUROG2 induction, it seems the cells by design are non-proliferative (anti-mitotic drug AraC is applied, among other strategies). But without a proliferative stem cell culture, how are independent experiments performed? Is the NEUROG2 induction performed multiple times on the same clone (a true biological replication) to yield 3 independent plates? Or are glutamatergic-induced cells frozen and plated at multiple times, another valid approach? Or alternatively, is a single induction plated at the same time to multiple plates? The issue of reproducibility is critical in this kind of study, and it is surprising that there is no mention of these issues that are currently central to the field.

Results:

Line 19-2 is used as a control for one family and is also the line for all isogenic lines produced. It is the only line shown that was reprogrammed by retrovirus (all other lines were induced using non-integrating Sendai virus vectors), so what is known about virus integration site mutations, and does this interfere or alternatively facilitate outcomes, and impact its use as a control? To make it easier for the reader to understand this report, the authors need to provide a description of line 19-2 here, rather than refer the reader to the un-reviewed manuscript in bioRxiv.

Supplementary file 2 indicates that for some families, only a single line was assessed for pluripotency markers and surprisingly, this seems to include 2 important control lines, 19-2 and NR3. It seems a concern that a control line used for many experiments was analyzed using only a single clone; please address.

The authors indicate they "sought to determine if any selected ASD-risk variants would interfere with spontaneous spiking and synchronized bursting activity in a whole network of interconnected glutamatergic neurons." To determine when to make MEA comparative analysis they pooled the N=341 control line wells from the 12 families, and observed that the mean firing rate (MFR) was the highest (~1.7 Hz) at week six (6) post-NEUROG2-induction (PNI). Thus they used this time point to compare all cell lines for the 1) weighted mean firing rate (MFR) and the 2) network burst frequency. This approach seems reasonable and seems to establish a standard set of conditions and protocols, one of their goals, as stated in the Abstract.

However, to this reviewer, the choice of a single time to make assessments seems to diminish the likelihood of finding differences between control and disease related genetic variants. I think it is not obvious nor expected that control lines from each family would exhibit the same maturation profile, and while the authors do perform intra-familial comparisons, the advantages of this type of analysis seem to be thwarted in this design. Given genetic heterogeneity, one could easily imagine different time frames. With all the expenses in cost, media, and time to set up these complex models, why not make recording every week for example? The current protocol as presented seems to be an inefficient use of resources. And perhaps in support of this criticism, the authors themselves seem to agree, as shown in Figure 3! Because the Family D mutant line #27 that carries *CNTN5*^+/-^ revealed differences in MEA at 6 weeks, they created an isogenic line 19-2 control line by inserting the *CNTN5* Stop-Tag. However they report they did not find any effects on MEA parameters at 6 weeks. Thus they extended analysis to 11 weeks, and now found statistically significant differences at 10 weeks. However, it seems that at week 7, the outcome appears to be the opposite, with StopTag showing 50% less activity (Figure 3B, left; would likely be significant with a t-test). It seems the authors cannot have it both ways, we use 6 weeks, but if we do not find an expected outcome, we wait longer. Further, maybe many of the other negative MEA results would appear different if assessed at other time points.

There is also concern that cell survival plays an important role. In *CNTN5*^+/-^, for example, perhaps cells with lower firing rates die off with progressive incubation. This would result in survival of only cells with increased MFR. However, because there is no report of the number of negative electrodes in each plate (which might increase with time), the results at 11 weeks appear to be the reverse of that seen at 7 weeks, yet this outcome may have little to do with synaptic functions. To this reviewer, since time plays an important role in genetic variant effects (Figure 3B), and the authors seem to tentatively conclude that changes in MEA measured activity depend on changes at the synapse, it would strengthen their case if single cell electrophysiology of these cells at this or an earlier time were correlated with the MEA outcomes, to demonstrate that *CNTN5* effects are mediated by changes in synaptic currents, and not in cell survival, distribution, or adhesion. Cellular analysis at MEA culture termination would address this unexplored mechanism.

The definition of weighted mean firing rate is not clear; the authors state, "We monitored the weighted mean firing rate (MFR), which represents the MFR per active electrode,.…". If understood correctly, this measure only averages active electrodes in any MEA well. Thus if there is significant (and currently un-described) failure of electrode activation in a well, that data is excluded. A major feature of a cell line may be that in one case only 20-30% of electrodes have signal due to differences in cell survival or dispersion, whereas in another line, it may routinely be 80-100% coverage. If this is the case, this type of information may impact the kinds of cellular mechanisms that one considers to explain the outcomes. This kind of information should be assessed, if not already done, and should be reported transparently in a Table that includes all families, perhaps in the supplement. One might find a correlation of decreasing/increasing cell-electrode contact (appearing as active/inactive electrodes/well) to the outcomes reported.

There is concern about the sole use of retrovirally-induced line 19-2 for all isogenic experiments. In contrast to control lines from the 10 families (Families C-L, Supplementary file 3), whose study led to selecting 6 weeks for MEA analysis, line 19-2 with introduced CRISPR gene variant only reveal differences at 8-12 weeks. There raises two issues. Firstly, MEA analysis of, for example, *EHMT2*^+/-^, demonstrates abnormal MFR and burst frequency only from weeks 4 to 7 when compared to family controls (Figure 5A), a result that is lost by week 8. In contrast, the 19-2 isogenic line only shows differences at week 9 or later. Similar differences are observed for *CNTN5*^+/-^ in Figure 3. Second, the MFR in line 19-2 is 4 Hz or more, whereas analysis of all the other 10 family control lines (Sendai based) examined at 6 weeks (sometimes 4-8 weeks) are mostly less than 1 Hz, and always below 2 Hz. These consistent differences raise concerns that the 19-2 line brings a special test system in which perhaps most of the analyzed variants could/should be assessed. But this may also bias results because only this single line is used for isogenic studies. Would this same effect occur in a Sendai non-integrating virus-induced iPSC model for isogenic line production?

Subsection “*EHMT2*^-/+^ CRISPR-Isogenic Pair Confirms Neuronal Hyperactivity”: The authors should reserve interpretations of differences in expression of MEA in the isogenic line compared to the family lines, 38B/E. They state: "This increased activity in mutant neurons occurred later than that observed in the familial lines 38B/E, possibly due to different genetic backgrounds." Alternatively, there are changes in cell survival, aggregation, electrode contact, or distribution that have not been assessed. It is premature to raise the issue of genetic background as explanation. Furthermore, the authors seem to be mistaken, reporting there is "lower resting membrane potential," but that is not what is shown in Figure S5 where the mean values appear identical; please correct this results error.

The discussion fails to mention any limitations to the approach of MEA at 6 weeks, the absence of cell characterization, and possible alternative mechanisms to the currently pre-conceived notions of the gene's function. The authors might in fact suggest that a full time course analysis would be beneficial. They fail to discuss the very different behavior of retrovirally created 19-2 line used at 9-12 weeks that show outcomes not observed in the Sendai virus created lines assessed at 6 weeks. They fail to address the possibility of explanatory mechanisms in their studies such as changes in cell survival, composition, distribution, or electrode contact. In all, these studies seem preliminary in nature and set a stage for further analysis before we recommend this specific set of procedures to be adopted by the field.

Additional data files and statistical comments:

A major feature of a cell line may be that in one case only 20-30% of electrodes have signal due to differences in cell survival or dispersion, whereas in another line, it may routinely be 80-100% coverage. If this is the case, this type of information may impact the kinds of cellular mechanisms that one considers to explain the outcomes. This kind of information should be assessed, if not already done, and should be reported transparently in a Table that includes all families, perhaps in the supplement. One might find a correlation of decreasing/increasing cell-electrode contact (appearing as active/inactive electrodes/well) to the outcomes reported.

[Editors’ note: further revisions were suggested before publication.]

The revised manuscript has been evaluated by a previous referee, who found the majority of the concerns have been adequately addressed by reorganizing the data and presenting the findings in a consistent manner. The Reviewing Editor's assessment is that several minor issues require attention. There is a need to acknowledge alternative explanations for the results and for further clarification of the MEA phenotypes. The editors feel that these issues can be dealt with by small revisions and additions to the text. Please make all the edits as outlined below.

Reviewer #1:

The authors have done a good job responding to my criticisms. The manuscript now more transparently details the variability between hiPSCs/experiments. The authors present phenotypic MEA and electrophysiological differences in CNTN5+/- and EHMT2+/- hiPSC-neurons from ASD. While they now clarify in the abstract that they identified synaptic phenotypes in 25% of the individuals tested, they should more clearly note that they cannot rule out synaptic phenotypes in the other 75% of ASD cases. For example, secondary differences in cellular replication, survival or patterning could mask activity differences by MEA. Moreover, the authors should consider in the discussion that MEA phenotypes need not predict electrophysiological deficits and vice versa, as evidenced for ASTN2 in their recent Stem Cell Reports publication. Finally, the authors still fail to resolve whether deficits in CNTN5 and EHMT2 result from impaired maturation rather than impaired synaptic function. With modest and mostly textual revisions, this report will be suitable for publication at *eLife*.

Minor Comments:

The nomenclature used in Figure 2–supplement 1 is very unclear. Can the authors add the relevant family ID and gene name such that the reader does not need to constantly refer back to Table 1?

---

## [Author Response]

Reviewer #1:

[…] Major detail:1) ASD neuronal phenotypes – The authors conducted MEA across all 25 hiPSC lines (Supplementary file 3) but do not seem to comprehensively present this data anywhere, focusing instead on CNTN5^-/+^ and EHMT2^-/+^. The findings (even if null) should be aggregated and clearly presented as a main figure. Additionally, the authors should make all raw MEA data and all hiPSC lines widely available upon publication.

We thank the reviewer for these helpful suggestions. Accordingly, we have aggregated all the null data (previously presented as Figures S2 and S3) along with the positive data into a single new Figure 3. All raw MEA data and all iPSC lines will be publicly available upon request at the time of publication. We are also open to submitting the iPSC cell lines to an appropriate repository if one is available. We are in discussions with Autism Speaks who have expressed interest in supporting this. As is the case for all of our bioresources described in publications, they will be available upon request. We adhere closely to standard academic publishing principles.

2) Inter-individual and intra-individual variability – The authors have generated perhaps the largest systematic MEA dataset across unrelated controls and ASD patients with different rare mutations. While they aggregate all control data in SI Figure 2 and 3, it would be tremendously valuable to the field if they might plot their results by donor as a main figure. Being able to show how MEA activity changes over time, and between donors, would greatly inform our understanding of inter-individual variation in MEA. Similarly, in those cases where the same line was tested on multiple MEA plates, again, it would be useful to present variability.

We agree that plotting all the results by donor as a main figure would help to appreciate the inter-individual and intra-individual variability. Hence, we have aggregated all the data in a new Figure 3 showing how the activity changes over time, and between donors, and even between different biological replicates for each donor. Correspondingly, we have added the following (framed) paragraphs to the section “Multi-Electrode Array Analysis of iPSC-Derived Neurons” of the Results:

“The weighted MFR (wMFR), which represents the MFR per active electrode, was used as a primary read-out for all tested iPSC-derived neurons, at one-week intervals from week 4 to 8 post-NGN2-induction (PNI) (Figure 3). To identify a preferred timepoint for this screen, we first pooled the data of all the independent control lines. Since the highest wMFR value for this pool of ‘all controls’ (~1.8 Hz) was detected at week 6 (Figure 3A), we initially used that timepoint to compare the activity of ASD variant and control lines for each family. In two different families, i.e., *CNTN5* and *EHMT2*, a significant higher wMFR was recorded in ASD variant neurons at week 6 compared with their corresponding familial control neurons (Figure 3B). *EHMT2* had a strikingly increased wMFR at all timepoints, whereas *CNTN5* at other timepoints was equivalent to its controls. We therefore ranked these two genes as high priorities for further study. In contrast, no significant differences were observed for DLGAP2, CAPRIN1, SET or GLI3 (Figure 3C), suggesting that these variants do not differ from control neuronal activity in our MEA assays, and therefore were not studied further. Different dynamics of altered wMFR were observed for *ANOS1* and VIP at week 4 (Figure 3D), and the *ANOS1* nonsense variant was ranked as an example of a candidate for further study. Conversely, a significant lower wMFR was recorded at weeks 7 and 8 for THRA (Figure 3D). No unaffected family members were available as controls for the single NR3 line (NRXN1) nor for 36O-36P (AGBL4), thus they were not chosen for further study. When we compared their values to the pooled values recorded from all the different familial controls available, no difference was found for NRXN1 and a significantly lower wMFR was observed at weeks 5 and 7 for AGBL4 (Figure 3E).

To explore intra-individual (different lines from the same individual) and inter-individual (different individuals with the same mutation) variability, we plotted all the values obtained from each single well, independent experiment, cell line and individual, at each of the five reading timepoints (Figure 3F). Most lines from an individual were not significantly different from each other, and reassuringly low inter-individual variability was observed with different siblings bearing the same mutation(s), e.g., 48K and 48N versus 49G and 49H (SET), or 61I and 61K versus 62M and 62X (GLI3), at different timepoints (Figure 3F). A few lines showed a significant intra-individual variability, e.g., lines 52A and 52C (THRA) at week 4, or lines 75G and 75H (CAPRIN1) at week 8 (Figure 3F). We also noted some inter-independent experiment variability for a given line, e.g., line 38E (*EHMT2*) at weeks 4 and 5 (dots with different colours do not overlap in Figure 3F). Note that similar profiles were monitored in terms of MFR (Figure 3—figure supplement 1), indicating that these differences were not due to having more or less active electrodes in different lines. While consistent activity across lines was generally observed, the presence of variability prompted us to interrogate independent variants created by genome editing of *CNTN5, ANOS1* and *EHMT2*.”

3) Inter-well and inter-hiPSC neuronal induction efficiency – It is incredibly difficult to assess and adjust for differences in total cell number and cell type composition in MEA experiments, both of which are expected to impact MEA recordings. What was the authors' strategy? It would be helpful if in the discussion they might inform the readers how best to learn from and extend their analyses.

The third and fourth paragraphs of the Discussion, as follow, now discuss this issue:

“We assume that most NGN2-neurons are glutamatergic based on the data presented in the original publication establishing this technique (Zhang et al., 2013), and on our previous publication using high-cell density RNAseq assessment (Deneault et al., 2018). Moreover, we have treated several of our cultures at the end of the MEA experimentation with different neurotoxins (CNQX, PTX, TTX) to show that most neurons are glutamatergic and not GABAergic, for different lines. In addition, our patch-clamp recordings have demonstrated that these neurons exhibit the properties of excitatory neurons.

Characterization of neuronal composition and survival when MEA is performed is difficult to achieve with high accuracy. Our strategy involved using several technical (3 to 12 per independent experiment) and biological (up to 4) replicates to best compensate for inter-well and inter-iPSC neuronal induction variations (Supplementary file 3). It is possible that some phenotypes were missed since we cannot exclude the possibility that differences in cell number or composition across individuals in a family actually masked potential phenotypes. However, this issue was ruled out for *CNTN5* and *EHMT2* because the phenotypes were confirmed in unrelated isogenic pairs, and by patch-clamp recordings.”

4) CRISPR-engineering – How did the authors rule out off-target effects following CRISPR-editing to produce isogenic pairs?

In our recent publication, which is now out in the peer-reviewed literature (Deneault et al., Stem Cell Reports, 2018), we have designed a new approach to detect any off-target mutations based on whole-genome sequencing of a series of isogenic KO lines. We have used the same strategy here and inserted the following (framed) sentence within the second paragraph of the “Repair of *ANOS1* Rescues Defective Membrane Currents” section of the Results:

“No overt off-target mutations were detectable using our previously-described WGS strategy (Deneault et al., 2018).”

This was expected since we used the double-nicking approach. Regarding the isogenic pairs made using ribonucleoprotein (RNP) complexes in line 19-2, i.e., 19-2-*CNTN5* and 19-2-*EHMT2*, we have not investigated any potential off-target mutations since this method was previously described with very low probability of off-target mutations by several different groups. Moreover, finding similar significant phenotypes as those observed in the patient lines further dissociated any potential off-target mutations to such phenotypes.

5) Phenotypic characterization – Increased neuronal activity might indicate alterations in synaptic function and/or maturation. For the two genes focused on within the manuscript, CNTN5^-/+^ or EHMT2^-/+^, the authors should provide attempt to resolve this question (even a time course of transcriptomic changes over 6-weeks of NGN2-induction).

We agree that an increased neuronal activity might indicate alterations in synaptic function and/or maturation. We now discuss this in the fifth paragraph of the Discussion, as following:

“An increased neuronal activity, e.g., MFR in mutant lines 38B/E, might indicate alterations in synaptic function and/or maturation. We have presented the MFR for all tested cell lines in Figure 3—figure supplement 1, in support of the wMFR in Figure 3. The wMFR is defined as the MFR divided by the number of active electrodes per well. If there is significant failure of electrode activation in a well, for example due to differences in cell survival, dispersion or adhesion, those data are excluded. For example, the increased MFR observed in *EHMT2*^-/+^ lines could be due to a better capacity to survive, disperse or adhere than *EHMT2*^+/+^ cells, without affecting synaptic activity. However, excluding all inactive electrodes would then result in a comparable wMFR between mutant and control cells, which was not the case. Indeed, both MFR and wMFR were significantly higher in *EHMT2*^-/+^ cells. That does not exclude the possibility of a better survival, dispersion or adhesion, but it is likely not the only reason for the observed increase in spiking activity, suggesting greater synaptic activity, as supported by patch-clamp recordings. Indeed, we used patch-clamp recordings to show that sEPSC frequency was significantly increased in mutant line 38E compared to control line 37E. We believe these results directly support synaptic alteration as one of the possible causes for the increased neuronal activity measured by MEA. However, a detailed analysis of cell maturation will be required for each different cell line involved in this study to clarify this issue.”

Moreover, as explained in the last paragraph of the Results, we also performed patch-clamp recordings on 19-2-*EHMT2* neurons at day 21-25 PNI. We did not detect any significant change in sEPSC frequency and amplitude at this earlier timepoint, similarly to the MEA experiment. However, intrinsic properties showed a significant increase in capacitance and decrease in input resistance in mutant cells (Figure 6—figure supplement 2). These observations suggest that the mutant neurons at 3-4 weeks PNI potentially have a faster maturation rate, however, this phenotype is most pronounced in the hyperactivity recorded by MEAs later at 9-11 weeks PNI. These results support the conclusion that the inactivation of one allele of *EHMT2* significantly increases spontaneous network activity of excitatory neurons, with possible effects on the neuronal maturation process. We also agree that a time course of RNAseq on the isogenic pairs could be informative regarding synaptic function and/or maturation pathways in the future. However, the presence of mouse astrocytes in the electrophysiological experiments could potentially limit the full understanding of what is happening at the two relevant timepoints.

6) Mechanism – Can the authors speculate as to the mechanism through which CNTN5^-/+^ or EHMT2^-/+^ increase neuronal activity? Possibly by linking their results to existing animal models? Perhaps RNAseq analyses might provide insights into the downstream changes in patient-derive neurons?

Speculation about potential mechanisms and linking to existing animal models are part of the Discussion. We have used RNAseq analyses in a previous publication (Deneault et al., 2018) to provide insights into the possible downstream changes in several different isogenic pairs involving KO of other ASD genes, and revealed interesting convergence of DEGs and networks. In the future, this tool will certainly help to shed light into the downstream genes and pathways affected by the absence of *EHMT2*, for example, which is mostly located in the nucleus and could be impacting activity-dependent transcription of synaptic signaling genes.

7) Data sharing – The authors have generated a great resource that is likely to be of great interest to the community. The value of such a collection would be enhanced by making publically available the inclusion of SNP and exome genotype data from each donor and clarification of the specific hiPSC lines available (at which repository). How many validated hiPSCs per donor are being made available? Through which repository? Will the MEA data be made available?

All raw MEA data, as well as all iPSC lines, will be publicly available upon request. Genotype data, in many different forms, from each donor and the relevant controls described in the paper is already available through the Autism Speaks MSSNG WGS project (Yuen et al., 2017). Of course, as our group always does, we will follow standard academic publishing rules and provide any information/resource needed for others to replicate our experiments.

Minor detail:1) Clarification of terminology – NGN2-overexpression is best described as "induction" not "differentiation." The authors are cautioned against describing the neurons as "cortical" without clear evidence of patterning.

We agree with this comment and we have made appropriate modifications throughout the manuscript where applicable.

2) Can the authors clarify in the main text (not just in the Materials and methods) that astrocyte co-culture was used with NGN2-neurons to improve maturation for MEA and electrophysiological.

We have modified the main text accordingly.

Additional data files and statistical comments:Please make available all MEA raw data to the community.

All raw MEA data will be publicly available upon request at the time of publication. We are also actively searching for a public repository to accept it (including the National Database for Autism Research).

Reviewer #2:

[…] Specific comments follow:Abstract: It seems to this reviewer that the Abstract communicates a more positive result than was actually achieved, as regards the "standardized set of procedures," that "used a large-scale multi-electrode array (MEA).…" These statements seem to suggest the value of this approach to the general field. Yet, of the 12 families with genetic variants, only 3 of the 12 MEA studies revealed any differences, which does not necessarily recommend this as a mainstay approach. The alternative approach, using single cell electrophysiology, defined only one additional pathology. The Abstract should be modified to change the emphasis, and consider stating the success rate of 33%.

We have modified the Abstract accordingly:

“We used a multi-electrode array, with patch-clamp recordings, to determine a synaptic phenotype in 25% of the individuals with ASD.”

Materials and methods:The authors do not mention performing a power analysis in designing the study, and power analysis is not discussed in the statistical methods. There is concern that the inherent variability in the MEA data is not well appreciated nor well controlled, and without this analysis, it cannot be transparently reported.

We agree that variability in the MEA data could not have been well appreciated in the previous version of the manuscript. In a two-sample comparison setting, a power analysis estimates the sample size required to likely detect an effect of a given size. In a screening approach, the sensitivity and specificity of the screen are also influenced by the sample size. Here, we elected to use MEA as a screening tool because we had previously evaluated the sensitivity of our protocol. For example, MEA was used to confirm a hypoactive neuronal phenotype, previously detected using patch-clamp recordings, in 4 out of 5 KO lines, in which 6-8 MEA wells were recorded per line per experiment, in mostly 4 independent experiments (Deneault et al., Stem Cell Reports, 2018). With such a high level of sensitivity, plus testing usually 2 different lines per individual, we assumed that aiming at 6 technical and 3 biological replicates per line would support enough sensitivity for our screen. In addition, we have aggregated all the data in a new Figure 3 showing how the activity changes over time, and between donors, and even between different biological replicates for each donor. Correspondingly, we added the following (framed) paragraphs to the section “Multi-Electrode Array Analysis of iPSC-Derived Neurons” of the Results:

“The weighted MFR (wMFR), which represents the MFR per active electrode, was used as a primary read-out for all tested iPSC-derived neurons, at one-week intervals from week 4 to 8 post-NGN2-induction (PNI) (Figure 3). To identify a preferred timepoint for this screen, we first pooled the data of all the independent control lines. Since the highest wMFR value for this pool of ‘all controls’ (~1.8 Hz) was detected at week 6 (Figure 3A), we initially used that timepoint to compare the activity of ASD variant and control lines for each family. In two different families, i.e., *CNTN5* and *EHMT2*, a significant higher wMFR was recorded in ASD variant neurons at week 6 compared with their corresponding familial control neurons (Figure 3B). *EHMT2* had a strikingly increased wMFR at all timepoints, whereas *CNTN5* at other timepoints was equivalent to its controls. We therefore ranked these two genes as high priorities for further study. In contrast, no significant differences were observed for DLGAP2, CAPRIN1, SET or GLI3 (Figure 3C), suggesting that these variants do not differ from control neuronal activity in our MEA assays, and therefore were not studied further. Different dynamics of altered wMFR were observed for *ANOS1* and VIP at week 4 (Figure 3D), and the *ANOS1* nonsense variant was ranked as an example of a candidate for further study. Conversely, a significant lower wMFR was recorded at weeks 7 and 8 for THRA (Figure 3D). No unaffected family members were available as controls for the single NR3 line (NRXN1) nor for 36O-36P (AGBL4), thus they were not chosen for further study. When we compared their values to the pooled values recorded from all the different familial controls available, no difference was found for NRXN1 and a significantly lower wMFR was observed at weeks 5 and 7 for AGBL4 (Figure 3E).

To explore intra-individual (different lines from the same individual) and inter-individual (different individuals with the same mutation) variability, we plotted all the values obtained from each single well, independent experiment, cell line and individual, at each of the five reading timepoints (Figure 3F). Most lines from an individual were not significantly different from each other, and reassuringly low inter-individual variability was observed with different siblings bearing the same mutation(s), e.g., 48K and 48N versus 49G and 49H (SET), or 61I and 61K versus 62M and 62X (GLI3), at different timepoints (Figure 3F). A few lines showed a significant intra-individual variability, e.g., lines 52A and 52C (THRA) at week 4, or lines 75G and 75H (CAPRIN1) at week 8 (Figure 3F). We also noted some inter-independent experiment variability for a given line, e.g., line 38E (*EHMT2*) at weeks 4 and 5 (dots with different colours do not overlap in Figure 3F). Note that similar profiles were monitored in terms of MFR (Figure 3—figure supplement 1), indicating that these differences were not due to having more or less active electrodes in different lines. While consistent activity across lines was generally observed, the presence of variability prompted us to interrogate independent variants created by genome editing of *CNTN5, ANOS1* and *EHMT2*”

There is no peer-reviewed, published characterization by the authors of their results of using NEUROG2 expression to induce glutamatergic neurons, characterizations that other cited authors have performed. While the authors state that; "Importantly, we determined that this strategy offers highly uniform differentiation levels between cell lines derived from different participants (Deneault et al., 2018);" that reference is to a manuscript submitted to bioRxiv that has not undergone peer review, and for the current article to be published, analysis of the NEUROG2 induction procedure needs to be presented. The statement that "differentiation levels" were uniform between cell lines, is quite a claim, and it is not at all clear what this means. For example what endpoints were assessed among multiple lines to indicate uniformity? One can imagine, for example, that failure to observe a specific differentiation feature, could reflect that the marker is not expressed by a cell, versus, that it is expressed but at a time point not examined. Such an outcome could reflect changes in i) cell fate, ii) induction of differentiation, iii) survival of a specific cell class, or simply, iv) a delay in its expression in suboptimal culture conditions. There is no characterization of the surviving cell population when MEA analysis of the cultures is performed, so that specific experimental outcomes cannot be related to the cell population under study. Even though the culture bottoms are opaque, immunocytochemistry using fluorescent labeling would allow visualization of cell numbers (DAPI) and cell type (NeuN, MAP2, vGlut, vGAD, etc) by use of the epifluorescence microscope to understand how cell composition at the time of MEA recording impacted the outcomes. This seems a reasonable request since the majority of MEA studies did not reveal group genetic differences, with only 3 of the 12 families demonstrating changes in firing rate and network burst frequencies.

We have used the NEUROG2 overexpression strategy for its high consistency in differentiation levels between several different lines. The reference to our bioRxiv paper was made for expediency and openness, and the paper has now undergone stringent and full peer-review and is published (Deneault et al., Stem Cell Reports, 2018). The third and fourth paragraphs of the Discussion, as followed, now discuss this issue:

“We assume that most NGN2-neurons are glutamatergic based on the data presented in the original publication establishing this technique (Zhang et al., 2013), and on our previous publication using high-cell density RNAseq assessment (Deneault et al., 2018). Moreover, we have treated several of our cultures at the end of the MEA experimentation with different neurotoxins (CNQX, PTX, TTX) to show that most neurons are glutamatergic and not GABAergic, for different lines. In addition, our patch-clamp recordings have demonstrated that these neurons exhibit the properties of excitatory neurons.

Characterization of neuronal composition and survival when MEA is performed is difficult to achieve with high accuracy. Our strategy involved using several technical (3 to 12 per independent experiment) and biological (up to 4) replicates to best compensate for inter-well and inter-iPSC neuronal induction variations (Supplementary file 3). It is possible that some phenotypes were missed since we cannot exclude the possibility that differences in cell number or composition across individuals in a family actually masked potential phenotypes. However, this issue was ruled out for *CNTN5* and *EHMT2* because the phenotypes were confirmed in unrelated isogenic pairs, and by patch-clamp recordings.”

The MEA methods are incomplete as reported, and extensive review of Supplementary file 3 still leaves uncertainties for this reviewer. The authors employ MEA as the single major measure of outcome, but they do not reference their previous use of this technology. It is not adequate to cite other laboratories. The authors should describe more fully the nature of the MEA technology, how the cultures appear in them, the data that is generated, and how it was analyzed. To start, the authors indicate that 48-well opaque-bottom MEA plates (Axion Biosystems) were used. But going to the website there are four (4) "Classic" varieties of this plate, as well as many other formats. Please provide exact catalogue number or numbers. How many electrodes are there in each well? It may be helpful to show a layout/picture for the electrode array, and also pictures of fixed and stained neuronal cultures to define the variability in cell contact to multiple electrodes, and perhaps the extremes of electrode contact they observe. Doing a brief PubMed search, I found two recent 2018 publications that provide images that show large variability of cell growth around fixed electrode arrays (DeRosa et al., Sci Rep 2018; Odawara et al., Sci Rep 2018; also see Hammond et al., BMC Neurosci, 2013). Please state the range of individual MEA wells used in these types of studies: Supplementary file 3 seems to indicate a range from 3 wells to as many as 12. How many independent MEA assay experiments are performed for each line? Supplementary file 3 is confusing, specifically the column labeled "Total nb independent plates." This number coincides with the reported number of different MEA plates (MEA plate IDs are 1-37, with only 21 involved here). Is an independent plate a separate experiment? Usually in iPSC studies, within a single clone (as reported here), independent experiments imply different passages of the same clone, analyzed in subsequent weeks, or after thawing an earlier passage and newly plated. But after NEUROG2 induction, it seems the cells by design are non-proliferative (anti-mitotic drug AraC is applied, among other strategies). But without a proliferative stem cell culture, how are independent experiments performed? Is the NEUROG2 induction performed multiple times on the same clone (a true biological replication) to yield 3 independent plates? Or are glutamatergic-induced cells frozen and plated at multiple times, another valid approach? Or alternatively, is a single induction plated at the same time to multiple plates? The issue of reproducibility is critical in this kind of study, and it is surprising that there is no mention of these issues that are currently central to the field.

We agree that the MEA technology is still in its infancy, but it is still important data to use, and in doing so we will help to move the field forward. We have now referenced our previous use of the MEA technology in the first sentence of the “Multi-Electrode Array Analysis of iPSC-Derived Neurons” section of the Results. Data generation and analysis is explained in Materials and methods. We have now provided the exact catalogue number of the MEA plates used, as well as the number of electrodes per well in Materials and methods. Since only opaque-bottom 48-well MEA plates were commercially available at the time this study was performed, it was not possible to take clear pictures of cell-electrode contact. We apologize for the confusion regarding Supplementary file 3, and have now modified it to reflect the reviewer’s requests. For example, we have replaced the column label "Total nb independent plates" for “Total nb independent experiments", as an independent plate meant independent experiment. Here, independent experiments imply independent infections with NGN2 viruses of iPSCs at different passages, entailing completely independent inductions. So, NGN2 induction was performed multiple times on the same clone (a true biological replication) to yield 3 independent plates. We have clarified this issue in the legend of Supplementary file 3.

Results:Line 19-2 is used as a control for one family and is also the line for all isogenic lines produced. It is the only line shown that was reprogrammed by retrovirus (all other lines were induced using non-integrating Sendai virus vectors), so what is known about virus integration site mutations, and does this interfere or alternatively facilitate outcomes, and impact its use as a control? To make it easier for the reader to understand this report, the authors need to provide a description of line 19-2 here, rather than refer the reader to the un-reviewed manuscript in bioRxiv.

To help us stay within length restrictions, the reader is now referred to our peer-reviewed and published paper (Deneault et al., 2018), where line 19-2 was extensively characterized.

Supplementary file 2 indicates that for some families, only a single line was assessed for pluripotency markers and surprisingly, this seems to include 2 important control lines, 19-2 and NR3. It seems a concern that a control line used for many experiments was analyzed using only a single clone; please address.

The purpose of using the line 19-2 here was to validate any finding in an independent genetic background while providing an isogenic control via CRISPR editing. Since two different clones, e.g., 38B and 38E, have replicated an increased neuronal activity (Figures 3 and 6), and that properly editing one clone requires extensive time and costs, we thought that one genetically-independent and isogenic clone would be sufficient to support our primary findings. We have tried to accompany all our findings by using conservative wording throughout the manuscript.

The authors indicate they "sought to determine if any selected ASD-risk variants would interfere with spontaneous spiking and synchronized bursting activity in a whole network of interconnected glutamatergic neurons." To determine when to make MEA comparative analysis they pooled the N=341 control line wells from the 12 families, and observed that the mean firing rate (MFR) was the highest (~1.7 Hz) at week six (6) post-NEUROG2-induction (PNI). Thus they used this time point to compare all cell lines for the 1) weighted mean firing rate (MFR) and the 2) network burst frequency. This approach seems reasonable and seems to establish a standard set of conditions and protocols, one of their goals, as stated in the Abstract.However, to this reviewer, the choice of a single time to make assessments seems to diminish the likelihood of finding differences between control and disease related genetic variants. I think it is not obvious nor expected that control lines from each family would exhibit the same maturation profile, and while the authors do perform intra-familial comparisons, the advantages of this type of analysis seem to be thwarted in this design. Given genetic heterogeneity, one could easily imagine different time frames. With all the expenses in cost, media, and time to set up these complex models, why not make recording every week for example? The current protocol as presented seems to be an inefficient use of resources. And perhaps in support of this criticism, the authors themselves seem to agree, as shown in Figure 3! Because the Family D mutant line #27 that carries CNTN5^+/-^ revealed differences in MEA at 6 weeks, they created an isogenic line 19-2 control line by inserting the CNTN5 Stop-Tag. However they report they did not find any effects on MEA parameters at 6 weeks. Thus they extended analysis to 11 weeks, and now found statistically significant differences at 10 weeks. However, it seems that at week 7, the outcome appears to be the opposite, with StopTag showing 50% less activity (Figure 3B, left; would likely be significant with a t-test). It seems the authors cannot have it both ways, we use 6 weeks, but if we do not find an expected outcome, we wait longer. Further, maybe many of the other negative MEA results would appear different if assessed at other time points.

We thank the reviewer for the constructive critique of our design. These are early days in performing this type of analysis and with standards of data presentation in place, we try to present the most relevant data to the experiments at hand. In the previous version of the manuscript, we assumed that it would be clearer for the reader if we show only one timepoint, i.e., the most active in control cells (week 6). However, as suggested by both reviewers, we decided to aggregate all the data and build a new Figure 3, showing how the activity changes over time, and between donors, and even between different biological replicates for each donor. Now, we think this is a better way to appreciate the variability of MEA, and consider potential difference in maturation rate of various lines. Line 19-2 is generally more active than most other familial lines, and this may have delayed emergence of the hyperactive phenotype in isogenic cells until week 10, as now discussed in the new version. Moreover, we have previously detected hypoactive phenotypes in line 19-2 at week 8 and before (Deneault *et al.*, Stem Cell Reports, 2018).

There is also concern that cell survival plays an important role. In CNTN5^+/-^, for example, perhaps cells with lower firing rates die off with progressive incubation. This would result in survival of only cells with increased MFR. However, because there is no report of the number of negative electrodes in each plate (which might increase with time), the results at 11 weeks appear to be the reverse of that seen at 7 weeks, yet this outcome may have little to do with synaptic functions. To this reviewer, since time plays an important role in genetic variant effects (Figure 3B), and the authors seem to tentatively conclude that changes in MEA measured activity depend on changes at the synapse, it would strengthen their case if single cell electrophysiology of these cells at this or an earlier time were correlated with the MEA outcomes, to demonstrate that CNTN5 effects are mediated by changes in synaptic currents, and not in cell survival, distribution, or adhesion. Cellular analysis at MEA culture termination would address this unexplored mechanism.

We agree that patch-clamp recordings of *CNTN5*^+/-^ neurons should help to support our MEA findings in the future. However, we have used weighted MFR (wMFR) (Figure 3), in addition to MFR (Figure 3—figure supplement 1), to take into account electrodes that might be inactive due to cell survival, distribution or adhesion.

The definition of weighted mean firing rate is not clear; the authors state, "We monitored the weighted mean firing rate (MFR), which represents the MFR per active electrode,.…". If understood correctly, this measure only averages active electrodes in any MEA well. Thus if there is significant (and currently un-described) failure of electrode activation in a well, that data is excluded. A major feature of a cell line may be that in one case only 20-30% of electrodes have signal due to differences in cell survival or dispersion, whereas in another line, it may routinely be 80-100% coverage. If this is the case, this type of information may impact the kinds of cellular mechanisms that one considers to explain the outcomes. This kind of information should be assessed, if not already done, and should be reported transparently in a Table that includes all families, perhaps in the supplement. One might find a correlation of decreasing/increasing cell-electrode contact (appearing as active/inactive electrodes/well) to the outcomes reported.

We apologize if the definition of wMFR was not made clear in the previous manuscript. It is defined as the MFR divided by the number of active electrodes per well. An electrode was considered active if it detected at least five spikes per minute, as explained in the section “Multi-electrode array (MEA)” of the Materials and methods. If there is significant failure of electrode activation in a well, for example, due to differences in cell survival, dispersion or adhesion, that data is indeed excluded. However, a detailed analysis of cell survival, dispersion and adhesion, will be required for each different cell line involved in this study to clarify this issue in the future. This is one of the reasons why we decided in the new version of the manuscript to present the MFR (Figure 3—figure supplement 1) in order to support the wMFR to take into account potential issue with cell survival, dispersion or adhesion. For example, the increased MFR observed in *EHMT2*^-/+^ lines in could be due to a better capacity to survive, disperse or adhere than *EHMT2*^+/+^ cells, without affecting synaptic activity. However, excluding all inactive electrodes would then result in a comparable wMFR, which is not the case (Figure 3B). Indeed, both MFR and wMFR are significantly higher in *EHMT2*^-/+^. That does not exclude the possibility of a better survival, dispersion or adhesion, but it is likely not the only reason for the observed increase in spiking activity, suggesting greater synaptic activity, as supported by patch-clamp recordings in Figure 6B. We now discuss this in the fifth paragraph of the Discussion.

There is concern about the sole use of retrovirally-induced line 19-2 for all isogenic experiments. In contrast to control lines from the 10 families (Families C-L, Supplementary file 3), whose study led to selecting 6 weeks for MEA analysis, line 19-2 with introduced CRISPR gene variant only reveal differences at 8-12 weeks. There raises two issues. Firstly, MEA analysis of, for example, EHMT2^+/-^, demonstrates abnormal MFR and burst frequency only from weeks 4 to 7 when compared to family controls (Figure 5A), a result that is lost by week 8. In contrast, the 19-2 isogenic line only shows differences at week 9 or later. Similar differences are observed for CNTN5^+/-^ in Figure 3. Second, the MFR in line 19-2 is 4 Hz or more, whereas analysis of all the other 10 family control lines (Sendai based) examined at 6 weeks (sometimes 4-8 weeks) are mostly less than 1 Hz, and always below 2 Hz. These consistent differences raise concerns that the 19-2 line brings a special test system in which perhaps most of the analyzed variants could/should be assessed. But this may also bias results because only this single line is used for isogenic studies. Would this same effect occur in a Sendai non-integrating virus-induced iPSC model for isogenic line production?

We agree that line 19-2 is generally more active than other tested lines in our hands, and we cannot exclude at this stage the possibility that it could be due to the use of retrovirus for reprogramming. However, we don’t think that these differences in intensity and time window should preclude the use of this line for validation/isogenic experiments. Indeed, we have previously shown that a completely independent line, i.e., 50B (Sendai-reprogrammed), was able to reproduce significant electrophysiological and RNAseq results obtained in isogenic KO studies performed in line 19-2 (Deneault et al., 2018). Line 19-2 has also been extensively used in other projects to validate different results obtained with familial lines defective in *SHANK2* (Zaslavsky *et al.*, Nature Neuroscience, accepted manuscript), and *PTCHD1-AS* (Ross *et al.*, Biological Psychiatry, in revision). Furthermore, a pattern of outward/inward currents comparable to that found in 18C/18CW lines (Figure 5D) was monitored in a different and unrelated isogenic pair, in which our StopTag fragment was previously inserted within the ANOS1 coding sequence in line 19-2 (Deneault et al., 2018). This result was found on the right panel of Figure 5D in the original manuscript. However, we prefer now to exclude this panel from the revised manuscript and put this profile only in the response to reviewers (see Author response image 1) since the trace of the control 19-2 was previously reported in (Deneault et al., 2018). This isogenic pair 19-2/19-2-*ANOS1*^StopTag/+^ phenocopies the genetically-independent isogenic pair 18C/18CW (Figure 5D). Since line 19-2 could be used to validate both hypoactive and hyperactive electrophysiological phenotypes, we don’t think that this line create any significant bias on its own.

**Author response image 1. respfig1:** Outward and inward membrane current detected by patch-clamp recordings; total number of recorded neurons was 20 for 19-2-ANOS^-/y^ and 33 for control 19-2; values are presented as mean+SEM of three independent differentiation experiments, recorded at day 21-25 PNI. *p < 0.05 from multiple t test comparison; ** note that control 19-2 profile was previously reported in Deneault et al., 2018

Subsection “EHMT2^-/+^ CRISPR-Isogenic Pair Confirms Neuronal Hyperactivity”: The authors should reserve interpretations of differences in expression of MEA in the isogenic line compared to the family lines, 38B/E. They state: "This increased activity in mutant neurons occurred later than that observed in the familial lines 38B/E, possibly due to different genetic backgrounds." Alternatively, there are changes in cell survival, aggregation, electrode contact, or distribution that have not been assessed. It is premature to raise the issue of genetic background as explanation. Furthermore, the authors seem to be mistaken, reporting there is "lower resting membrane potential," but that is not what is shown in Figure S5 where the mean values appear identical; please correct this results error.

We agree that it is probably premature to raise the issue of genetic background as explanation. Accordingly, we have modified “possibly due to different genetic backgrounds" for “possibly due to the more active 19-2 line”. As now explained earlier in the second paragraph of the “*CNTN5* Isogenic Pair to Control for Genetic Background Contribution” section of the Results: “However, the wMFR of line 19-2 increased up to nearly 3 Hz at week 8 (Figure 4B) while the *CNTN5* family controls stayed around 0.5 Hz (Figure 3B). In this context of a more active cell line, we extended the recordings until week 11, and the hyperactive wMFR of 19-2-*CNTN5*^StopTag/+^ was only evident from week 10 (Figure 4B).”

Moreover, based on our response to one of the other points raised above by this reviewer, we think that the issue of cell survival and electrode contact is not a major contributor to the different time window observed in the activity in lines 38B/E and 19-2. We apologize for the confusion regarding the lower “resting membrane potential”, we have now changed it for lower “input resistance” in the main text.

The discussion fails to mention any limitations to the approach of MEA at 6 weeks, the absence of cell characterization, and possible alternative mechanisms to the currently pre-conceived notions of the gene's function. The authors might in fact suggest that a full time course analysis would be beneficial. They fail to discuss the very different behavior of retrovirally created 19-2 line used at 9-12 weeks that show outcomes not observed in the Sendai virus created lines assessed at 6 weeks. They fail to address the possibility of explanatory mechanisms in their studies such as changes in cell survival, composition, distribution, or electrode contact. In all, these studies seem preliminary in nature and set a stage for further analysis before we recommend this specific set of procedures to be adopted by the field.

In the new version of the manuscript, we present a time course of the wMFR and MFR from week 4 to 8 PNI, which addresses the reviewer’s concern about the use of only week 6 as a timepoint. We now discuss the different behavior of line 19-2 at the end of the second paragraph in the Discussion. Moreover, we discuss some limitations of using MEA as a primary read-out, e.g., “It is possible that some phenotypes were missed since we cannot exclude the possibility that differences in cell number or composition across individuals in a family actually masked potential phenotypes.” We also discuss that since the familial controls are often not very active, this screen might be biased towards the identification of hyperactive phenotypes rather than hypoactive. Better MEA controls will need to be developed in the future.

Additional data files and statistical comments:A major feature of a cell line may be that in one case only 20-30% of electrodes have signal due to differences in cell survival or dispersion, whereas in another line, it may routinely be 80-100% coverage. If this is the case, this type of information may impact the kinds of cellular mechanisms that one considers to explain the outcomes. This kind of information should be assessed, if not already done, and should be reported transparently in a Table that includes all families, perhaps in the supplement. One might find a correlation of decreasing/increasing cell-electrode contact (appearing as active/inactive electrodes/well) to the outcomes reported.

This point was addressed above.

[Editors’ note: further revisions were suggested before publication.]

· We added a few new words to the Abstract (it now slightly exceeds 150 words). This additional text was added to address the reviewer's suggestion about the 75% of lines without detectable phenotypes. If you need to have the Abstract at 150 words please just remove the additions (the previous presentation was fine with us too);

· We made a few new modifications to the 4th paragraph of the Discussion to respond to the reviewer's first two requests;

· We clarified the nomenclature of Figure 2–figure supplement 1, and modified the corresponding legend;

· We modified the last box of the transparent reporting file to include the new Source data files that were recently uploaded;

· We brought more precision on the type of multiple t-test comparisons that were used in the legend of Figures 3-6;

· Regarding the last request of the reviewer, we acknowledge that we did not completely resolve whether deficits (or phenotypes) in CNTN5 and EHMT2 result from impaired maturation rather than impaired synaptic function (we will never be able to do everything), however, we feel that we have already explained this extensively in the 5th paragraph of the Discussion.